# Offline Opponent Modeling with Truncated Q-driven Instant Policy Refinement

Yuheng Jing [1 2]  Kai Li [† 1 2]  Bingyun Liu [1 2]  Ziwen Zhang [1 2]  Haobo Fu [3]  Qiang Fu [3]  Junliang Xing [4]
Jian Cheng [† 1 5 6]

## Abstract

**O**ffline **O**pponent **M**odeling (**OOM**) aims to learn an adaptive autonomous agent policy that dynamically adapts to opponents using an offline dataset from multi-agent games. Previous work assumes that the dataset is optimal. However, this assumption is difficult to satisfy in the real world. When the dataset is suboptimal, existing approaches struggle to work. To tackle this issue, we propose a simple and general algorithmic improvement framework, **T**runcated Q-driven **I**nstant **P**olicy **R**efinement (**TIPR**), to handle the suboptimality of OOM algorithms induced by datasets. The TIPR framework is plug-and-play in nature. Compared to original OOM algorithms, it requires only two extra steps: (1) Learn a horizon-truncated in-context action-value function, namely Truncated Q, using the offline dataset. The Truncated Q estimates the expected return within a fixed, truncated horizon and is conditioned on opponent information. (2) Use the learned Truncated Q to instantly decide whether to perform policy refinement and to generate policy after refinement during testing. Theoretically, we analyze the rationale of Truncated Q from the perspective of No Maximization Bias probability. Empirically, we conduct extensive comparison and ablation experiments in four representative competitive environments. TIPR effectively improves various OOM algorithms pretrained with suboptimal datasets.

## 1. Introduction

A fundamental task towards Artificial General Intelligence is the development of *autonomous agents* capable of modeling others. The line of work that focuses on adversarial domains is commonly referred to as **O**pponent **M**odeling (**OM**), where we build a *self-agent* (*i.e.*, *autonomous agent*) that models the behaviors, goals, or other properties of *opponents* to reduce its uncertainty about the environment and enhance its decision-making (He et al., 2016a; Albrecht & Stone, 2018; Papoudakis et al., 2021a; Nashed & Zilberstein, 2022; Fu et al., 2022; Yu et al., 2022; Ma et al., 2024; Jing et al., 2024b; 2025). Among the research, **O**ffline **OM** (**OOM**) is a recently proposed learning paradigm (Jing et al., 2024a). OOM aims to learn an adaptive self-agent policy that can dynamically adapt to opponents based on the available opponent information using offline datasets. This new paradigm makes OM more practical and efficient, as it relaxes the reliance on interactions with the environment and opponent policies in the learning process.

Existing work on OOM assumes that the pre-collected multi-agent game dataset is ***optimal*** (Jing et al., 2024a). Here, 'optimal' means that for each opponent policy in the offline dataset, all trajectories are generated by the opponent policy and its *Best Response* (BR) self-agent policy. However, the experiences we can collect are often ***suboptimal***, making it difficult to meet the above optimality requirement. By 'suboptimal', we mean that within the dataset, the self-agent policy can be arbitrarily bad rather than being the BR to the opponent policy. As expected, we observe that when pretrained on suboptimal datasets, existing OOM algorithms deteriorate dramatically. This contradicts the original intention of OOM, which is to improve learning efficiency through offline learning. Regarding this, we aim to explore how to make OOM effective using suboptimal datasets in this work. The concept of OOM with suboptimal data is reflected in many real-world applications. *E.g.*, NBA players study opponents' styles and strategies by analyzing game replays to identify their weaknesses, even though many replays include videos of losses against those opponents.

In Offline *Reinforcement Learning* (RL), a mainstream methodoloy to acquire a policy potentially better than the one embedded in the dataset is to learn an additional

---

[†]Corresponding authors  [1]C²DL, Institute of Automation, Chinese Academy of Sciences  [2]School of Artificial Intelligence, University of Chinese Academy of Sciences  [3]Tencent AI Lab  [4]Tsinghua University  [5]School of Future Technology, University of Chinese Academy of Sciences  [6]AiRiA. Correspondence to: Kai Li <kai.li@ia.ac.cn>, Jian Cheng <jian.cheng@ia.ac.cn>.

*Proceedings of the $42^{nd}$ International Conference on Machine Learning*, Vancouver, Canada. PMLR 267, 2025. Copyright 2025 by the author(s).

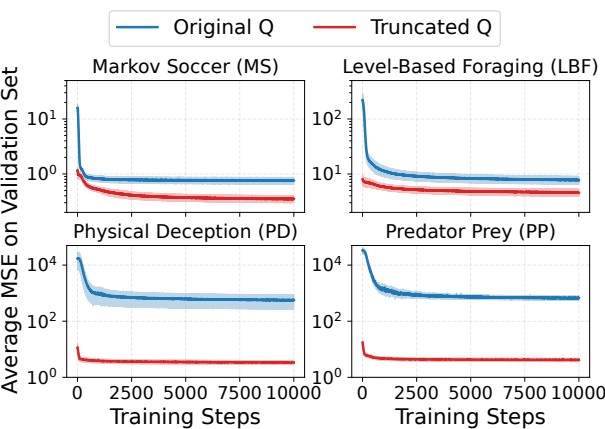

Figure 1. The *Mean Squared Error* (MSE) curves on the validation sets for learning different Q functions with various suboptimal offline datasets from different environments.

action-value function Q for *Offline Conservative Learning* (OCL) (Wu et al., 2019; Fujimoto et al., 2019; Kumar et al., 2020; Jin et al., 2021). Inspired by OCL, an intuitive way to improve the original policy learned by OOM algorithms from suboptimal datasets is to refine it through Q. However, in the context of OOM, learning a workable Q is highly challenging. On the one hand, the involvement of opponents introduces additional action dimensions to Q, adding extra estimation complexity. On the other hand, the *non-stationarity* of opponents (*i.e.*, *opponents switching policies*) makes Q' estimation unreliable.

For the first challenge, we find that shortening the horizon over which Q estimates the expected return can significantly reduce the learning difficulty. Fig. 1 shows the error curves of learning Q on validation datasets. The error of learning the **Original Q**, which estimates the expected *cumulative discounted reward* (*i.e.*, *return*) over the full horizon, is unacceptably large , especially in environments with more opponents, *e.g.*, PD and PP. In contrast, learning the **Truncated Q**, which estimates the expected return over a truncated horizon, results in order-of-magnitude reduction in error. For the second challenge, we argue that learning an *in-context* Q, *conditioned on opponent information*, is more beneficial for improving the reliability of the estimation under the current opponent policy than an unconditional Q.

In addition to how to learn Q, how to use Q also matters in OOM. The previously mentioned OCL emphasizes learning an improved policy on the offline trajectory distribution. However, under OOM's setting, opponents during testing can be entirely unseen, leading to severe distributional shifts. This can significantly degrade the performance of OCL on the test trajectory distribution. Considering this, we propose a novel method, **Instant Policy Refinement** (**IPR**), to improve the original policy of OOM algorithms during testing.

In summary, we propose a simple and general algorithmic improvement framework, **T**runcated Q-driven **I**nstant **P**olicy **R**efinement (**TIPR**), to enhance OOM algorithms in handling suboptimality induced by datasets. The TIPR framework is plug-and-play and requires only two extra steps compared to original OOM algorithms: (1) ***Truncated Q Training***: Learn a horizon-truncated in-context action-value function, namely Truncated Q, using the same dataset used for pretraining the OOM algorithm. (2) ***IPR***: Use the learned Truncated Q to instantly decide whether to refine the original policy and generate the refined policy during testing.

Theoretically, we prove from the perspective of maximizing the No Maximization Bias probability that, compared to Original Q, our Truncated Q is potentially a more reasonable Q function. Empirically, we conduct extensive comparative and ablation experiments in four representative competitive environments. We construct offline datasets with varying degrees of suboptimality to pretrain a series of OOM algorithms and then test their performance against unknown non-stationary opponents. Our results demonstrate that under the TIPR framework, various OOM algorithms consistently achieve considerable performance improvements, regardless of the degree of suboptimality in the dataset.

## 2. Preliminaries

We use an $n$-agent Partially-Observable Stochastic Game $\langle \mathbb{S}, \{\mathbb{O}^i\}_{i=1}^n, \{\mathbb{A}^i\}_{i=1}^n, \mathcal{P}, \{R^i\}_{i=1}^n, \{\mathcal{O}^i\}_{i=1}^n, T, \gamma \rangle$ to formalize the multi-agent environment (Yang & Wang, 2020). Here, $\mathbb{S}$ denotes the state space. $\mathbb{O}^i$ denotes the observation space of agent $i \in [n]$. $\mathbb{A}^i$ denotes agent $i$'s action space, $\mathbb{A} = \prod_{i=1}^n \mathbb{A}^i$ is the joint action space. $\mathcal{P} : \mathbb{S} \times \mathbb{A} \times \mathbb{S} \to [0, 1]$ is the transition probabilities. $R^i : \mathbb{S} \times \mathbb{A} \to \mathbb{R}$ denotes the reward function of agent $i$. $\mathcal{O}^i : \mathbb{S} \times \mathbb{A} \times \mathbb{O}^i \to [0, 1]$ denotes the agent $i$'s observation function. $T$ is the full horizon for each game episode. $\gamma$ is the discount factor.

We use the superscript 1 to denote terms related to the *self-agent*, *i.e.*, *the agent under our control*, and the superscript $-1$ to denote terms related to the *opponents*, *i.e.*, *all other agents*. Both 1 and $-1$ represent terms in the joint space. The offline dataset is defined as $\mathbb{T} := \cup_k \mathbb{T}^k$, $\mathbb{T}^k := \{\tau^k := (o_t^{1,k}, o_t^{-1,k}, a_t^{1,k}, a_t^{-1,k}, r_t^{1,k}, r_t^{-1,k})_{t=0}^{T-1}\}$. The *set of offline opponent policies* embedded in $\mathbb{T}$ is denoted as $\Pi^{\text{off}} := \{\pi^{-1,k}\}_{k=1}^K$, where $k \in [K]$ is the index of the opponent policy. $\mathbb{T}^k$ contains multiple trajectories $\tau^k$ that result from the interaction between the opponent policy $\pi^{-1,k}$ and certain self-agent policies. Existing work assumes that for any $k \in [K]$, the actions $a^{1,k}$ generated by the self-agent policy are optimal (Jing et al., 2024a). In this work, we make no such assumptions about the self-agent policies embedded in $\mathbb{T}$, and they can be arbitrarily suboptimal.

In OOM, the data used to characterize opponent policies is

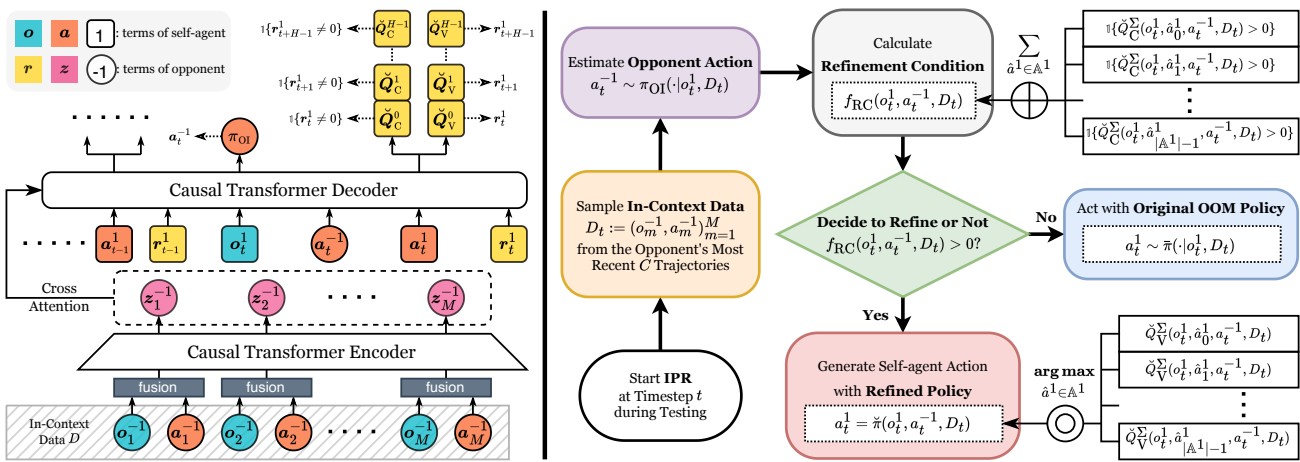

*Figure 2. Left*: Neural network and learning process of **Truncated Q**. The network of Truncated Q is primarily composed of an Encoder and a Decoder. The Encoder encodes the *In-Context Data* $D$, which is used to characterize the opponent's policy, into a latent variable $z$. This latent variable establishes a connection with the self-agent's input through cross-attention. The Decoder autoregressively uses $\breve{Q}_{\text{C}}$ and $\breve{Q}_{\text{V}}$ at each timestep to predict the **C**onfidence and **V**alue of rewards to be obtained over the next $H$ steps. *Right*: Overall procedure of **I**nstant **P**olicy **R**efinement (**IPR**). At each timestep during testing, IPR first samples $D$ from the opponent's recent trajectories and predicts the opponent's action using *Opponent Imitator* $\pi_{\text{OI}}$, then decides whether the *Refinement Condition* (RC) is met based on the prediction of $\breve{Q}_{\text{C}}$. If the RC is satisfied, IPR generates the *refined policy* $\breve{\pi}$ using $\breve{Q}_{\text{V}}$; otherwise, IPR maintains the *original OOM policy* $\bar{\pi}$.

typically called *In-Context Data* $D$. A straightforward way to construct $D$ is to sample several $(o^{-1}, a^{-1})$ tuples from the trajectory of a given opponent policy. By this way, the *set of D within* $\mathbb{T}$ can be defined as $\mathbb{D} := \cup_k \mathbb{D}^k$, $\mathbb{D}^k := \{D := (o_m^{-1}, a_m^{-1})_{m=1}^M | (o^{-1}, a^{-1}) \sim \tau^k \in \mathbb{T}^k\}$. The adaptive self-agent policy to be learned can be expressed as $\pi(a^1|o^1, D)$, which dynamically adjusts based on $D$ to adapt to the opponent. Given an opponent policy $\pi^{-1}$, the action-value function of self-agent can be defined as $Q_{\pi, \pi^{-1}}(o^1, a^1, a^{-1}) = \mathbb{E}_{\pi, \pi^{-1}} \left[ \sum_{t=0}^{T-1} \gamma^t r_t^1 \right]$. The objective of OOM is to pretrain $\pi$ using the dataset $\mathbb{T}$ so that it maximizes its expected action value when tested against a *set of online opponent policies* $\Pi^{\text{on}}$, *i.e.*, $\max_\pi \mathbb{E}_{\pi \leftarrow \text{OOMPretrain}(\mathbb{T}), \pi^{-1} \sim \Pi^{\text{on}}} Q_{\pi, \pi^{-1}}(\cdot, \cdot, \cdot)$. [1]

In Sec. A, we provide extensive related work on *Opponent Modeling*, *In-Context Learning*, *Offline RL*, *Offline Multi-Agent RL*, and *Transformers for Decision-Making*.

## 3. Methodology

OOM algorithms typically follow a supervised pretraining paradigm, and their performance heavily depends on the self-agent policies embedded in $\mathbb{T}$. However, in the real world, the collected $\mathbb{T}$ is often suboptimal, where the embedded self-agent policies can be arbitrarily poor. In such cases, the pretrained OOM algorithms could also be arbitrarily suboptimal. In this work, we propose a simple and general algorithmic improvement framework, **T**runcated Q-driven **I**nstant **P**olicy **R**efinement (**TIPR**). The framework

---
[1] $\pi, Q$ are self-agent's terms. Omit superscript 1 for simplicity.

learns a horizon-truncated in-context action-value function, *i.e.*, **Truncated Q** (see Sec. 3.1), to perform **I**nstant **P**olicy **R**efinement (**IPR**) during testing, thereby improving the pretrained OOM algorithm (see Sec. 3.2). Truncated Q is theoretically proven to have the advantage of maximizing the No Maximization Bias probability (see Sec. 3.3). Our TIPR is designed as a plug-and-play framework to address the suboptimality induced by $\mathbb{T}$ for OOM algorithms. The overview of our TIPR framework is illustrated in Fig. 2. We also provide the corresponding pseudocode in Algo. 1.

### 3.1. Truncated Q Training

Inspired by *Offline Conservative Learning* (OCL), which learns Q for conservative policy improvement to improve the policies embedded in datasets, we aim to learn a Q to improve OOM algorithms pretrained on suboptimal datasets. However, under the problem setting of OOM, learning a workable Q is highly challenging due to the following reasons: (1) *Challenge 1*: The multi-agent games involve opponents, and an accurate Q requires modeling the opponents' actions $a^{-1}$. This inherently introduces extra complexity to the learning of Q. (2) *Challenge 2*: During testing, opponents are non-stationary, meaning they can switch policies. In such cases, it is difficult to ensure the reliability of the Q estimation, as the current Q estimation may not correspond to the action values under the opponents' newest policy.

Under the formalization of OOM, assuming $\bar{Q}$ denotes the **Original Q** parameterized by $\omega$. Then, $\bar{Q}$ can be learned by:

$$\min_\omega \mathbb{E}_{(o_t^1, a_t^1, a_t^{-1}) \sim \tau \in \mathbb{T}} \left[ (\bar{Q}(o_t^1, a_t^1, a_t^{-1}) - G_t^1)^2 \right]. \quad (1)$$

Here, $G_t^1 = \sum_{t'=t}^{T-1} \gamma^{t'-t} r_{t'}^1$ represents the self-agent's full horizon *Return-To-Go* (RTG). From Eq. (1), we can intuitively observe that the error in the optimization objective of $\bar{Q}$ accumulates primarily through the accumulation of rewards over time $t'$ in $G_t^1$. Therefore, for a given sample $(o_t^1, a_t^1, a_t^{-1})$, the larger the absolute value of $T-t$, the more difficult it becomes for $\bar{Q}$ to fit that RTG label $G_t^1$.

As mentioned in **Challenge 1**, inclusion of opponent agents further exacerbates the difficulty of learning $\bar{Q}$. This effect becomes more pronounced as the number of opponents increases, as demonstrated in Fig. 1 for PD (2 opponents) and PP (3 opponents). To address this challenge, we argue that truncating the horizon over which Q estimates the expected return from the full length to a fixed, truncated length can mitigate the cumulative error effect during the learning of Q. This truncation makes it feasible to learn a workable Q, as thoroughly validated by experimental results in Sec. 4.2.

To address the aforementioned **Challenge 2**, we argue that a sound Q should estimate the action values *under the current opponent policy*. Such a Q can provide reliable estimations even in the presence of non-stationary opponents. Therefore, we design Q to be in-context, incorporating opponents' $(o^{-1}, a^{-1})$ as additional conditional inputs. This design potentially enables Q to characterize the current opponent policy through *In-Context Learning* (ICL) (Lin et al., 2024).

To sum up, we propose learning a horizon-truncated in-context action-value function **Truncated Q**, denoted as $\breve{Q}$, to overcome the challenges associated with learning Q in OOM. Assuming that $\breve{Q}$ estimates the expected return over a truncated horizon of $H$, $\breve{Q}$ decomposes the prediction of the expected return within $H$ steps into predictions of the specific rewards for each timestep. Here, $\breve{Q}$ is composed of $\breve{Q}_C$ and $\breve{Q}_V$. For a specific future timestep $h$, $\breve{Q}_C^h$ and $\breve{Q}_V^h$ predict the *Confidence of obtaining the $h$-th reward* and the *Value of the $h$-th reward*, respectively (see the *Left* side of Fig. 2). Specifically, we learn $\breve{Q}$ through the objective:

$$\min_{\omega} \mathbb{E}_{\substack{k\sim[K],(o_t^1,a_t^1,a_t^{-1},\{r_{t'}^1\}_{t'=t}^{t+H-1}) \\ \sim\tau^k\in\mathbb{T}^k, D\sim\mathbb{D}^k}} [\alpha \cdot \mathcal{L}_{Q_C} + \beta \cdot \mathcal{L}_{Q_V}].$$

Here, $\alpha$ and $\beta$ are the coefficients for different loss terms, and $D$ represents the In-Context Data mentioned in Sec. 2. We define the *confidence loss* $\mathcal{L}_{Q_C}$ for learning $\breve{Q}_C$ with *Binary Cross Entropy* (BCE) as follows:

$$\mathcal{L}_{Q_C} := \frac{1}{H} \sum_{h=0}^{H-1} \mathcal{L}_{\text{BCE}}(\breve{Q}_C^h(o_t^1, a_t^1, a_t^{-1}, D), \mathbb{1}\{r_{t+h}^1 \neq 0\}),$$

where $\mathcal{L}_{\text{BCE}}(x,y) = -[y\log(x)+(1-y)\log(1-x)]$, $\mathbb{1}\{\cdot\}$ is the indicator function, and the outputs of $\breve{Q}_C^h$ are assumed to have been processed by Sigmoid function $\sigma(x) = 1/(1+$

$\exp(-x))$. We define the *value loss* $\mathcal{L}_{Q_V}$ for learning $\breve{Q}_V$ using *Mean Square Error* (MSE) as follows:

$$\mathcal{L}_{Q_V} := \frac{1}{H} \sum_{h=0}^{H-1} (\breve{Q}_V^h(o_t^1, a_t^1, a_t^{-1}, D) - r_{t+h}^1)^2.$$

To maximize Truncated Q's ability to recognize opponents through ICL, our neural architecture adopts a causal Transformer (Radford et al., 2019). See more details on neural network design in Sec. E. Moreover, we introduce an auxiliary *Opponent Imitator* $\pi_{\text{OI}}(a^{-1}|o^1, D)$, which learns to estimate the opponents' action $a^{-1}$ given $o^1$ through conditional imitation learning, for the use during the IPR process.

### 3.2. Instant Policy Refinement

Learning a good Q is undoubtedly important, but how to use Q to improve OOM algorithms is equally critical. A direct way is to adopt the OCL methodology, using Q for policy improvement while adding conservative constraints to ensure the iteration does not deviate too far from the trajectory distribution of $\mathbb{T}$. Existing OOM algorithms typically adopt supervised pretraining objectives, where these objectives can conveniently serve as conservative constraints.

Assuming *original policy of OOM algorithm* is denoted as $\bar{\pi}$ and parameterized by $\theta$, the objective of learning a improved $\bar{\pi}$ using **OCL** based on $\breve{Q}$ can be informally written as:

$$\max_{\theta} \mathbb{E}_{\substack{k\sim[K],(o_t^1,a_t^1,a_t^{-1},\{r_{t'}^1\}_{t'=t}^{t+H-1}) \\ \sim\tau^k\in\mathbb{T}^k, D\sim\mathbb{D}^k, \hat{a}_t^1\sim\bar{\pi}(\cdot|o_t^1,D)}} \left[ \lambda \cdot \breve{Q}_V^\Sigma(o_t^1, \hat{a}_t^1, a_t^{-1}, D) - \eta \cdot \mathcal{L}_{\text{OOM}} \right].$$

Here, $\breve{Q}_V^\Sigma(\cdot,\cdot,\cdot,\cdot) := \sum_{h=0}^{H-1} \gamma^h \breve{Q}_V^h(\cdot,\cdot,\cdot,\cdot)$ denotes the *truncated cumulative value*. $\mathcal{L}_{\text{OOM}}$ typically adopts a cross-entropy form to imitate the actions of self-agent sampled from $\mathbb{T}$, which is also equivalent to certain Kullback-Leibler divergence constraint that prevents deviation from the trajectory distribution of $\mathbb{T}$ (Jing et al., 2024a). $\lambda$ and $\eta$ are the coefficients for the policy improvement term and the conservative constraint term, respectively.

The above OCL method emphasizes improving the policy on the trajectory distribution of $\mathbb{T}$ during offline training. However, in the context of OOM, opponents during testing are unknown and may even be entirely unseen. In such cases, severe distributional shifts can occur, rendering the improvements made offline ineffective on the new trajectory distribution. Therefore, rather than directly following OCL, we propose improving the original policy of the pretrained OOM algorithm during testing through IPR (see the *Right* side of Fig. 2). During testing, IPR first determines whether to refine the original policy based on the estimated confidence. If refinement is deemed necessary, IPR then generates the refined policy based on the estimated value.

We provide the detailed procedure of **IPR** as follows.

**Algorithm 1** **T**runcated Q-driven **I**nstant **P**olicy **R**efinement

**Require:** Offline dataset $\mathbb{T}$, original OOM policy $\bar{\pi}$ pretrained on $\mathbb{T}$, truncation horizon $H$

**Ensure:** IPR generated policy $\pi_{\text{IPR}}$

1:  /\* **Truncated Q Training (Section 3.1)** \*/
2:  Initialize **Truncated Q** model $\breve{Q}$'s parameters $\omega$
3:  **while** Training is not finished **do**
4:      Sample a batch of opponent policy index $k \sim [K]$
5:      Sample a batch of training data and label $(o_t^1, a_t^1, a_t^{-1}, \{r_{t'}^1\}_{t'=t}^{t+H-1}) \sim \tau^k \in \mathbb{T}^k$
6:      Sample a batch of *In-Context Data* $D \sim \mathbb{D}^k \sim \mathbb{T}^k$
7:      Compute *confidence loss* $\mathcal{L}_{Q_{\text{C}}}$, *value loss* $\mathcal{L}_{Q_{\text{V}}}$, and *imitation loss* $\mathcal{L}_{\text{OI}}$ for *Opponent Imitator* $\pi_{\text{OI}}$
8:      Update $\omega$ using gradients of $\alpha \cdot \mathcal{L}_{Q_{\text{C}}} + \beta \cdot \mathcal{L}_{Q_{\text{V}}} + \mathcal{L}_{\text{OI}}$
9:  **end while**
10:  /\* **Instant Policy Refinement (Section 3.2)** \*/
11:  **for** each timestep $t$ during testing **do**
12:      Get self-agent observation $o_t^1$
13:      Sample In-Context Data $D_t$ from the most recent $C$ opponent trajectories
14:      Estimate opponent actions with Opponent Imitator: $a_t^{-1} \sim \pi_{\text{OI}}(\cdot|o_t^1, D_t)$
15:      Compute *Refinement Condition* (RC) $f_{\text{RC}}$ by Eq. (2)
16:      **if** $f_{\text{RC}}(o_t^1, a_t^{-1}, D_t) > 0$ **then**
17:          Generate *refined policy* $\breve{\pi}$ by Eq. (3)
18:          Act using refined policy: $a_t^1 \leftarrow \breve{\pi}(o_t^1, a_t^{-1}, D_t)$
19:      **else**
20:          Act using *original OOM policy*: $a_t^1 \sim \bar{\pi}(\cdot|o_t^1, D_t)$
21:      **end if**
22:  **end for**
23:  **return** IPR generated policy $\pi_{\text{IPR}}$

---

**Prepare $D$ & $a^{-1}$.** At each timestep $t$, some preparations need to be made. First, we randomly sample $M$ tuples of $(o^{-1}, a^{-1})$ from the most recent $C$ historical trajectories of the opponent to construct In-Context Data $D_t$, which helps characterize the current opponent policy. Next, we use the Opponent Imitator $\pi_{\text{OI}}$ to sample $a_t^{-1} \sim \pi_{\text{OI}}(\cdot|o_t^1, D_t)$, as the opponents' action at timestep $t$ is unknown. After these preparations, we use the estimated confidence $\breve{Q}_{\text{C}}$ to determine whether to perform policy refinement.

**Decide whether to refine or not.** We define the *truncated cumulative confidence* as $\breve{Q}_{\text{C}}^{\Sigma}(\cdot, \cdot, \cdot, \cdot) := \sum_{h=0}^{H-1} Bern(\breve{Q}_{\text{C}}^h(\cdot, \cdot, \cdot, \cdot))$, where $Bern(\cdot)$ denotes sampling 1 or 0 from the Bernoulli distribution. Next, we define the *Refinement Condition* (RC) as follows:

$$f_{\text{RC}}(o_t^1, a_t^{-1}, D_t) := \sum_{\hat{a}^1 \in \mathbb{A}^1} \mathbb{1}\{\breve{Q}_{\text{C}}^{\Sigma}(o_t^1, \hat{a}^1, a_t^{-1}, D_t) > 0\}. \tag{2}$$

If $f_{\text{RC}}(\cdot, \cdot, \cdot) > 0$ (RC is satisfied), IPR performs policy refinement. Otherwise, IPR uses the *original OOM policy*

to generate self-agent action, *i.e.*, $a_t^1 \sim \bar{\pi}(\cdot|o_t^1, D_t)$.

**Generate the refined policy.** If the RC is satisfied, IPR use the estimated value $\breve{Q}_{\text{V}}$ to derive the *refined policy* $\breve{\pi}$. Specifically, $\breve{\pi}$ generates self-agent action $a_t^1$ by

$$\breve{\pi}(o_t^1, a_t^{-1}, D_t) := \arg\max_{\hat{a}^1 \in \mathbb{A}^1} \breve{Q}_{\text{V}}^{\Sigma}(o_t^1, \hat{a}^1, a_t^{-1}, D_t). \tag{3}$$

To summarize, at timestep $t$ during testing, given $o_t^1$, the prepared $D_t$ and $a_t^{-1}$, the self-agent policy induced by the IPR can be expressed as:

$$\pi_{\text{IPR}}(o_t^1, a_t^{-1}, D_t) := \begin{cases} \breve{\pi}(o_t^1, a_t^{-1}, D_t), & \text{RC is satisfied} \\ \bar{\pi}(\cdot|o_t^1, D_t), & \text{otherwise} \end{cases}.$$

Intuitively, IPR *performs policy refinement* (greedy) only when it has high confidence in the current value estimation, while *maintaining the original policy* (conservative) in other cases. This policy balancing mechanism is validated in our experiments to effectively trade-off between greediness and conservativeness, achieving better performance improvements compared to always being greedy or conservative.

### 3.3. The Rationale behind Truncated Q

Truncated Q shortens the horizon for estimating the expected return from the full length to a fixed, truncated length. This truncation's impact on the effectiveness of Q function requires further investigation. This subsection provides a theoretical analysis to justify the rationality of Truncated Q.

When using Q to improve the original policy, what really matters is ***whether the learned Q can correctly choose the action with the highest true expected return***. This is the most fundamental criterion for evaluating the effectiveness of the learned Q. Let the truncated horizon random variable be denoted as $\mathsf{h}$, and the neural network of Truncated Q be denoted as $\breve{Q}_{\mathsf{h}}$. Given any $o^1$, $a^{-1}$, and $D$, we aim for the following *Overall No Maximization Bias (NMB) Probability* $y(\mathsf{h})$ to be as large as possible:

$$y(\mathsf{h}) := P(\arg\max_{a^1} \breve{Q}_{\mathsf{h}} = \arg\max_{a^1} \mathbb{E}\mathsf{G}_T), \tag{4}$$

$\mathsf{G}_{\mathsf{h}}$ denotes the horizon-truncated RTG random variable, while $\mathsf{G}_T$ denotes the original RTG random variable. It is straightforward to derive that $y(\mathsf{h})$ satisfies that:

$$y(\mathsf{h}) \geq f(\mathsf{h})g(\mathsf{h}),$$
$$\text{where } f(\mathsf{h}) := P(\arg\max_{a^1} \breve{Q}_{\mathsf{h}} = \arg\max_{a^1} \mathbb{E}\mathsf{G}_{\mathsf{h}}), \tag{5}$$
$$g(\mathsf{h}) := P(\arg\max_{a^1} \mathbb{E}\mathsf{G}_{\mathsf{h}} = \arg\max_{a^1} \mathbb{E}\mathsf{G}_T).$$

Here, $f(\mathsf{h})$ denotes the *Empirical Risk NMB Probability*, which is determined by the neural network's fitting process. $g(\mathsf{h})$ denotes the *Natural NMB Probability*, which is determined by the intrinsic properties of the environment's reward structure. Next, we present the following theorem:

**Theorem 3.1.** *For any given* $o^1, a^{-1}, D$, *we have* $f(\mathsf{h}) \geq \mathfrak{f}(\mathsf{h}) := 1 - 2\eta_0$, *where* $\eta_0$ *satisfies that*

$$\Delta(\mathsf{h})\left(\sqrt{\frac{S(\ln \frac{2|\mathbb{T}|}{S} + 1) - \ln \eta_0/4|\mathbb{O}^1||\mathbb{A}||\mathcal{D}|}{l_{min}} + \frac{1}{l_{min}}} + \sqrt{\frac{-\ln \eta_0/|\mathbb{O}^1||\mathbb{A}||\mathcal{D}|}{2l_{min}}}\right) = \frac{U^2}{4}. \quad (6)$$

*The asymptotic complexity of the term* $\Delta(\mathsf{h})$ *is* $O(\mathsf{h}^2)$.

To facilitate a clear understanding of the rationale behind our theory, we provide the detailed explanations of all the notations used in Thm. 3.1 in Sec. B.1. The complete proof of Thm. 3.1 is in Sec. B.2. From Thm. 3.1, we conclude that as the number of opponents increases (leading to larger $|\mathbb{A}|$ and $|\mathcal{D}|$, where $\mathcal{D}$ is the set of all possible $D$), $\mathfrak{f}(\mathsf{h})$ decreases. This implies that fitting $\mathbb{E}\mathsf{G}_\mathsf{h}$ with $\check{Q}_\mathsf{h}$ becomes more challenging, which supports **Challenge 1** mentioned in Sec. 3.1 and aligns with the results shown in Fig. 1. Based on Thm. 3.1, we propose the following proposition:

**Proposition 3.2.** *For any given* $o^1, a^{-1}, D$, *there exists an optimal truncated horizon* $\mathsf{h}^* \in [T]$ *that maximizes* $\mathfrak{f}(\mathsf{h})g(\mathsf{h})$, i.e., *the lower bound of* $y(\mathsf{h})$. *Specifically, there exist functions* $g$ *such that* $\mathsf{h}^* \neq T$.

The proof and intuitive analysis of Prop. 3.2 can be found in Sec. B.3. According to Prop. 3.2, as $\mathsf{h}$ increases, $\mathfrak{f}(\mathsf{h})$ exhibits a decreasing trend, while $g(\mathsf{h})$ generally tends to increase. Therefore, selecting an appropriate truncated horizon $\mathsf{h}$ for estimating the expected return enables us to trade off between $\mathfrak{f}(\mathsf{h})$ and $g(\mathsf{h})$ to maximize $y(\mathsf{h})$. This, in turn, allows us to select the action with the highest $\mathbb{E}\mathsf{G}_T$ with higher probability when improving the original policy. To sum up, our theoretical analysis indicates that compared to unthinkingly setting $\mathsf{h}$ to $T$ (which is the case for Original Q), learning a Truncated Q is more reasonable.

## 4. Experiments

In this section, Sec. 4.1 provides a detailed description of the experimental setup. Sec. 4.2 poses a series of questions and presents empirical results to answer them, aiming to analyze the effectiveness of the TIPR framework.

### 4.1. Experimental Setup

**Environments.** We consider four sparse-reward competitive multi-agent environmental benchmarks. See Sec. C for detailed introductions of these environments.

- `Markov Soccer` (`MS`) is a two-player zero-sum game with a discrete state space. In `MS`, the self-agent's objective is to move the ball towards the opponent's goal, with the opponent having the same objective. The ball ran-

domly appears on the field at the beginning. `MS` requires players to be flexible in both offense and defense.
- `Level-Based Foraging` (`LBF`) is a two-player mixed-incentive game with a discrete state space. In `LBF`, the self-agent aims to eat as many apples as possible. All players and apples have a level. `LBF` represents a typical social dilemma and necessitates cooperation with the opponent to eat apples of a higher level than the self-agent's.
- `Physical Deception` (`PD`) is a three-player (two opponents) game with a continuous state space. Self-agent aims to hit an unknown target landmark, where there is a fake landmark and a target one. Opponents aim to prevent self-agent from hitting the target. Self-agent has to detect the opponents' deception and identify the real target.
- `Predator Prey` (`PP`) is a four-player (three opponents) game with a continuous state space. Self-agent is a prey aims to avoid being captured by three predators (opponents). There are two obstacles. The challenge of `PP` lies in the need to model all three opponents simultaneously and handle potential cooperation among them.

**OOM Baselines.** Considering that most existing OM approaches follow an online learning paradigm, we include OOM algorithms derived from pretraining-focused OM approaches adopted by Jing et al. (2024a) as baselines.

- **DRON-concat** (He et al., 2016a): Encode hand-crafted features of opponents with a linear network. It concatenates the self-agent's and the opponent's hidden states to aid the downstream policy optimization.
- **DRON-MoE** (He et al., 2016a): Encode hand-crafted features of opponents with a Mixture-of-Expert network while also predicting opponents' actions to model the opponents (the most performant version in their paper).
- **LIAM** (Papoudakis et al., 2021a): Use the observations and actions of the self-agent to reconstruct those of the opponent through an auto-encoder, thereby embedding the opponent policy into a latent space to model opponents.
- **Prompt-DT** (Xu et al., 2022): Based on Decision Transformer (Chen et al., 2021), sample expert task trajectories as prompts to obtain task adaptability through offline pretraining. We adopt the version in Jing et al. (2024a) that demonstrates great potential to handle OOM problems.
- **TAO** (Jing et al., 2024a): 1) Pretrain a well-structured opponent policy embedding with representation learning. 2) Learn to respond to the opponent policies based on the learned policy embedding with a Transformer model. 3) Adapt to unknown opponents with ICL during testing.

**Opponent Policies & Offline Datasets.** We employ a diversity-driven *Population-Based Training* algorithm MEP (Zhao et al., 2023) to train a policy population. Policies from this population are used to construct $\Pi^{off}$ and $\Pi^{on}$. Opponent policies generated using MEP have been shown to be performant and exhibit diversity (Jing et al., 2024b).

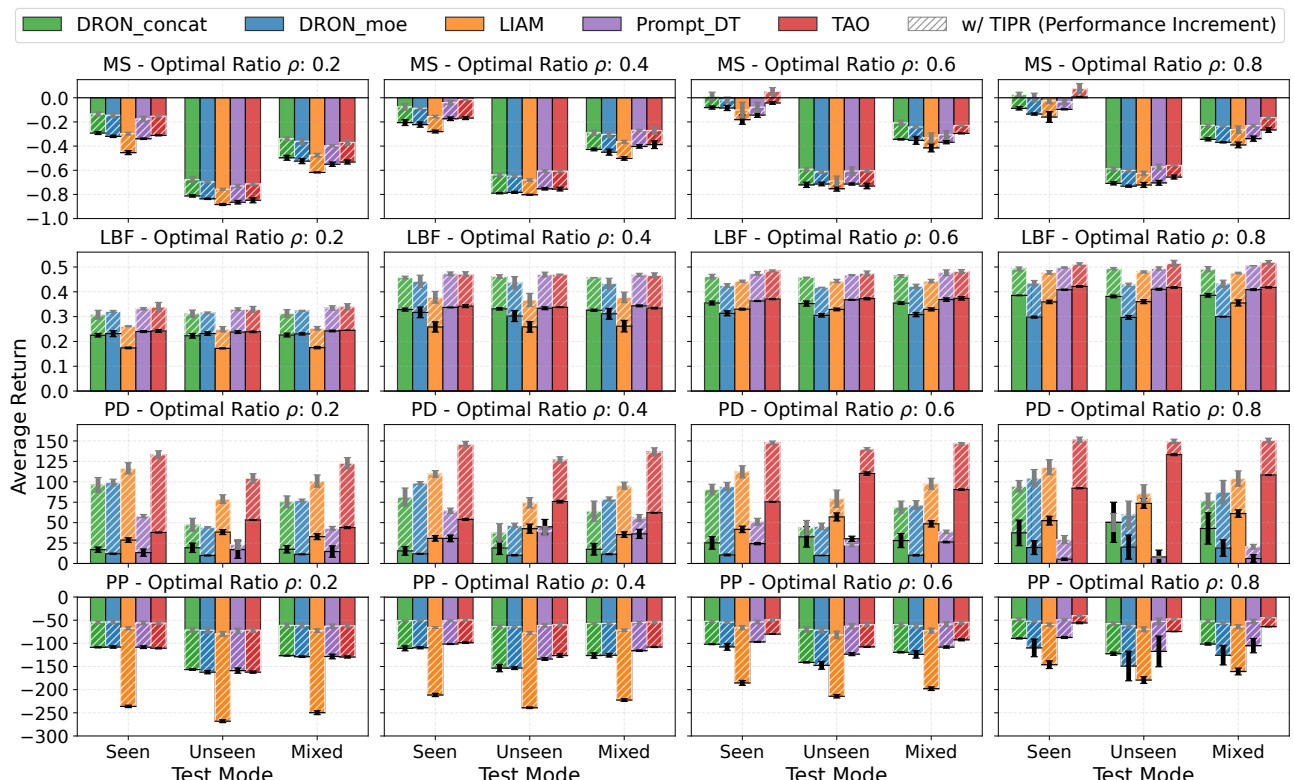

*Figure 3.* The average original results of various OOM baselines tested against unknown non-stationary opponents, along with the average results after improvement using the TIPR framework. The different subplots use various environments and offline datasets with different levels of suboptimality to pretrain the OOM algorithms. We use '**black error bars**' to represent the *Standard Deviation* (SD) of the original OOM policy's performance and '**grey error bars**' to represent the SD of the performance after applying TIPR.

We quantitatively measure and visualize the diversity of opponent policies within the MEP population used to construct the offline datasets $\mathbb{T}$ in Sec. D.

Offline datasets $\mathbb{T}$ with varying suboptimality, measured by the *Optimal Ratio* $\rho$, are constructed. We define $\rho$ as the ratio of the performance of self-agent policy embedded in $\mathbb{T}$ to the performance of the BR policy against the opponent policy. Self-agent policies used to construct $\mathbb{T}$ of different $\rho$ are obtained by training with PPO (Schulman et al., 2017) for varying numbers of steps while keeping opponent policy fixed. The smaller $\rho$, the more suboptimal the dataset $\mathbb{T}$ is.

**OOM Pretraining & Testing Protocols.** We pretrain all OOM baselines for 3000 steps. The final checkpoints of the pretrained OOM baselines were used to test against *Unknown Non-stationary Opponents* for 2400 episodes. 'Unknown' indicates that the true policy of the opponents is unknowable to the self-agent. 'Non-stationary' means that the opponent switches its policy by sampling from $\Pi^{on}$ every 20 episodes. We set up three types of $\Pi^{on}$:

1) Seen: This $\Pi^{on}$ is equivalent to $\Pi^{off}$, which contains 12 policies selected from the MEP population.
2) Unseen: This $\Pi^{on}$ contains 8 policies selected from the

MEP population that have never appeared in $\Pi^{off}$.
3) Mixed: This $\Pi^{on}$ is the union of the Seen and Unseen.

All figures and tables report the *Mean* and *Standard Deviation* (SD) of the results averaged over 5 random seeds. See all the hyperparameters in Sec. F.

### 4.2. Empirical Analysis

**Question 1.** *Can TIPR effectively handle the suboptimality of OOM algorithms induced by the offline datasets?*

Fig. 3 shows the original testing results of all OOM baselines and their results after applying the TIPR framework. We use a white shading pattern to highlight the performance increments achieved using the TIPR framework relative to the original OOM policy. It can be observed that as $\rho$ decreases, the performance of all OOM baselines generally drops dramatically, reaching an unacceptable level (especially in PD and PP). After applying the TIPR framework, all OOM baselines achieved consistent performance improvements. The improvements achieved by TIPR remain stable across different $\rho$ settings and are particularly pronounced in environments such as PD. These results demonstrate that TIPR can effectively handle the suboptimality of OOM algorithms

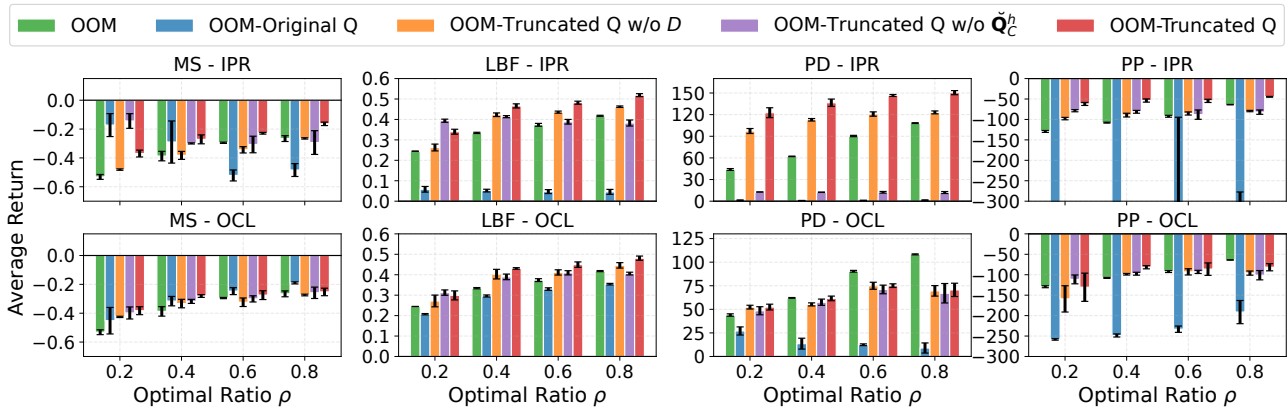

*Figure 4.* The average testing results of improving OOM algorithm using different variants of Q through both IPR and OCL methods, where the $\Pi^{\mathrm{on}}$ is set to Mixed. Some bars have extreme values, making them difficult to observe clearly. *E.g.*, in subplot 'PD - IPR', the average returns of 'OOM-Original Q' are close to 0. In subplot 'PP - IPR', the average returns of 'OOM-Original Q' are less than $-400$.

*Table 1.* The quantitative metrics of estimated values when improving OOM algorithms during testing using Original Q and Truncated Q through the IPR method, where the $\Pi^{\mathrm{on}}$ is set to Mixed.

| Env. | Evaluation Term | Optimal Ratio $\rho$ | | | |
|---|---|---|---|---|---|
| | | 0.2 | 0.4 | 0.6 | 0.8 |
| MS | $\bar{Q}$ MSE $\downarrow$ | $63.9 \pm 18.7$ | $97.5 \pm 52.4$ | $35.9 \pm 7.3$ | $70.5 \pm 14.1$ |
| | $\breve{Q}_{\mathrm{V}}$ MSE $\downarrow$ | $0.5 \pm 0.1$ | $0.8 \pm 0.2$ | $0.7 \pm 0.2$ | $0.7 \pm 0.1$ |
| | $\breve{Q}_{\mathrm{C}}$ Acc. (%) $\uparrow$ | $99.3 \pm 0.2$ | $99.0 \pm 0.2$ | $99.2 \pm 0.2$ | $99.3 \pm 0.1$ |
| LBF | $\bar{Q}$ MSE $\downarrow$ | $46.1 \pm 28.1$ | $23.6 \pm 1.5$ | $29.8 \pm 9.7$ | $42.0 \pm 14.0$ |
| | $\breve{Q}_{\mathrm{V}}$ MSE $\downarrow$ | $5.8 \pm 1.1$ | $5.1 \pm 1.2$ | $5.1 \pm 1.6$ | $4.7 \pm 1.6$ |
| | $\breve{Q}_{\mathrm{C}}$ Acc. (%) $\uparrow$ | $97.0 \pm 0.1$ | $97.4 \pm 0.6$ | $96.5 \pm 1.0$ | $97.1 \pm 1.4$ |
| PD | $\bar{Q}$ MSE $\downarrow$ | $42.8 \pm 9.3$ | $77.8 \pm 34.4$ | $253.9 \pm 54.7$ | $372.1 \pm 105.0$ |
| | $\breve{Q}_{\mathrm{V}}$ MSE $\downarrow$ | $2.9 \pm 0.1$ | $2.5 \pm 0.4$ | $2.6 \pm 0.3$ | $2.4 \pm 0.3$ |
| | $\breve{Q}_{\mathrm{C}}$ Acc. (%) $\uparrow$ | $90.7 \pm 10.7$ | $86.1 \pm 3.3$ | $88.9 \pm 6.3$ | $87.6 \pm 4.4$ |
| PP | $\bar{Q}$ MSE $\downarrow$ | $4.1e5 \pm 3.5e5$ | $3.2e5 \pm 3.5e4$ | $1.8e5 \pm 2.1e5$ | $3.1e5 \pm 2.5e5$ |
| | $\breve{Q}_{\mathrm{V}}$ MSE $\downarrow$ | $3.3 \pm 0.4$ | $2.9 \pm 0.2$ | $2.5 \pm 0.3$ | $2.1 \pm 0.3$ |
| | $\breve{Q}_{\mathrm{C}}$ Acc. (%) $\uparrow$ | $89.3 \pm 0.2$ | $90.2 \pm 0.7$ | $88.7 \pm 0.2$ | $90.9 \pm 1.1$ |

induced by the offline datasets.

**Question 2.** *Compared to OCL, can our proposed IPR more effectively improve OOM algorithms?*

In Fig. 4, we present the testing results of improving the policy of OOM algorithms using different Q variants through both IPR and OCL. We select the most representative OOM algorithm, TAO, for observation and analysis.[2] Let us focus on the results of improving OOM algorithm using Truncated Q through IPR and OCL, respectively. We observe that, compared to OCL, the IPR method generally provides a more effective improvement to the original OOM policy. OCL can even degrade the performance of OOM algorithms in environments such as PD. This can be attributed to the severe distribution shift between the trajectory distribution during testing and $\mathbb{T}$. As a result, performing improvements during testing is better suited to adapt to the new trajectory

---

[2] Due to space limits, subsequent ablations also focus on TAO.

distribution than making improvements offline.

**Question 3.** *Compared to Original Q, can using Truncated Q more effectively improve OOM algorithms?*

Let us continue observing Fig. 4. Focusing on the results of using Truncated Q and Original Q for improvements, we find: Regardless of whether the improvement is achieved through IPR or OCL, 'OOM-Truncated Q' generally outperforms 'OOM-Original Q' in effectively improving the policy. 'OOM-Original Q' can completely spoil the original OOM policy in many cases, such as in PD and PP.

Table 1 presents the statistics for predictions made when improving OOM algorithms using Original Q and Truncated Q through IPR during testing. We use *MSE* to measure the Q's value estimations and *Accuracy* to evaluate the precision of the Q's confidence. Truncated Q consistently demonstrates higher accuracy in its estimations across all games. In contrast, Original Q generally exhibits very low prediction accuracy and fails to produce meaningful predictions in PP. These observations support that Truncated Q is more effective than Original Q in improving OOM algorithms.

**Question 4.** *Do all the key design choices of Truncated Q contribute positively to improving OOM algorithms?*

Fig. 4 also includes the testing results of improving OOM algorithm using ablated variants of Truncated Q. Within, 'Truncated Q w/o $\breve{Q}_{\mathrm{C}}^{h}$' is a variant that removes the estimation of confidence, thus always performing policy refinement (greedy). 'Truncated Q w/o $D$' indicates the variant where the in-context data input is removed. Focusing on the results of improvements made using IPR, we can find that both 'Truncated Q w/o $\breve{Q}_{\mathrm{C}}^{h}$' and 'Truncated Q w/o $D$' generally suffer varying degrees of performance degradation compared to 'Truncated Q'. These observations demonstrate

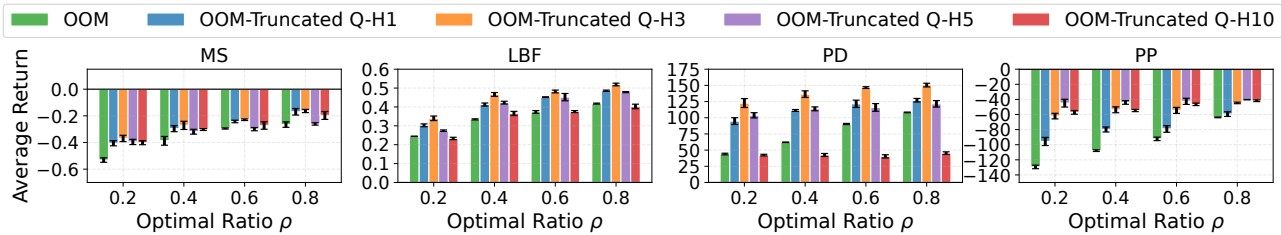

*Figure 5.* The average testing results of improving OOM algorithms using Truncated Q with different truncated horizons $H$, where the $\Pi^{on}$ is set to Mixed. We use different colors to represent Truncated Q trained with different values of $H$.

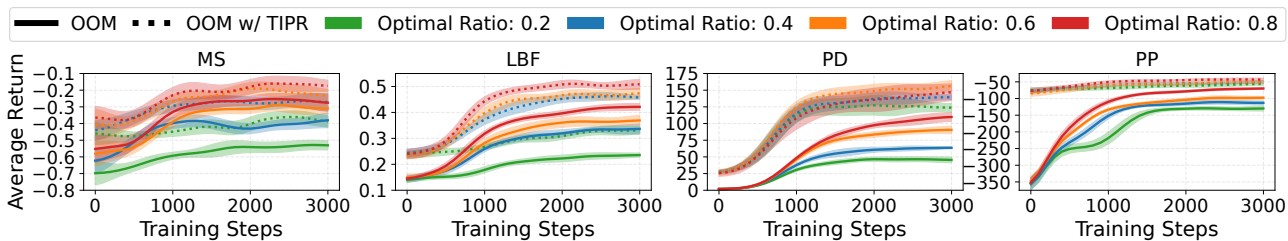

*Figure 6.* The average testing results of improving OOM algorithms during their pretraining stage using the TIPR framework, where the $\Pi^{on}$ is set to Mixed. We use different colors to represent the datasets with varying Optimal Ratios used for pretraining.

that weighing the decision to refine based on confidence estimation and characterizing opponent policies with $D$ both contribute positively to the effectiveness of Truncated Q.

**Question 5.** *How does the choice of different truncated horizons $H$ for Truncated Q affect the improvement results?*

In Fig. 5, we present the testing results of improving OOM algorithm using Truncated Q learned with different truncated horizons $H$ under the TIPR framework. We set four different values for $H$: 1, 3, 5, and 10. We observe that TIPR's improvement on OOM algorithm does not exhibit a strict correlation trend with the size of $H$. In different environments, an optimal value of $H$ exists, which can be considered a tunable parameter. However, when the value of $H$ becomes too large, it may lead to adverse improvements on the original OOM policy (*e.g.*, in LBF and PD). When $H = T$, Truncated Q degenerates into Original Q.

**Question 6.** *Can TIPR potentially improve the pretraining efficiency of OOM algorithms?*

Fig. 6 shows the results of using TIPR to improve OOM algorithms during their pretraining process. Specifically, we periodically take checkpoints during the OOM algorithm's pretraining stage for testing. We find that using TIPR during pretraining consistently and effectively improves OOM algorithms, regardless of the $\rho$ value of the $\mathbb{T}$ used for pretraining. Moreover, the improvements achieved by TIPR remain relatively stable from the beginning to the end of pretraining. This suggests that TIPR can enhance the pretraining efficiency of OOM algorithms. When $\mathbb{T}$ is suboptimal, using

TIPR can reduce the number of training steps required for OOM algorithms to achieve the same level of performance.

## 5. Discussion

**Summary.** This paper investigates a critical yet underexplored problem: How to address the degradation in algorithm performance caused by the suboptimality of offline datasets in OOM. We propose TIPR, a simple and general algorithmic improvement framework that learns a horizon-truncated in-context action value Truncated Q and performs IPR during testing to address this problem. Our theory justifies the rationality of Truncated Q by analyzing how the truncated horizon affects the No Maximization Bias probability. Experimental results demonstrate that TIPR effectively improves various OOM algorithms pretrained on suboptimal datasets, significantly enhancing their adaptability to unknown non-stationary opponents.

**Limitations and future work.** In Poiani et al. (2023), trajectories of different lengths are truncated to estimate the action value Q. Unlike our work, their goal is to find the optimal combination of truncation lengths under a given sampling budget to estimate Q as accurately as possible. Enlightened by their work, we think that, given a dataset, how to find the optimal truncated horizon $H$ for Truncated Q is a problem worth further investigation. Furthermore, in this work, the testing opponents choose a policy from a fixed set of policies for switching. An interesting future research direction would be exploring how to leverage offline datasets to handle opponents continuously updating their policies.

## Acknowledgements

This work is supported in part by the Strategic Priority Research Program of the Chinese Academy of Sciences (Grant No. XDA0480200), the National Science and Technology Major Project (2022ZD0116401), the Natural Science Foundation of China (Grant Nos. 62222606 and 61902402), and the Key Research and Development Program of Jiangsu Province (Grant No. BE2023016).

## Impact Statement

This work advances the field of OOM by addressing the critical limitation of suboptimal offline datasets, which are prevalent in real-world multi-agent scenarios. The proposed TIPR framework enhances the robustness and adaptability of autonomous agents without requiring additional online interactions, thereby reducing resource consumption and potential risks in deployment. Ethically, the framework encourages safer AI deployment by minimizing reliance on idealized data and promoting reliability under imperfect conditions. However, care must be taken to prevent misuse in adversarial or deceptive applications, particularly in competitive or high-stakes domains such as finance or security. Future societal implications include improved collaboration and competition among AI agents in diverse settings, fostering broader trust and utility in autonomous systems.

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

## A. Related Work

**Opponent Modeling (OM).**   Most prior OM work relies on a fully online learning paradigm, directly interacting with the environment and the opponent's policy rather than learning an opponent model from an offline dataset. The existing learning paradigms generally boil down to two stages: (1) *Pretraining*: pretrain an opponent model with designed OM methodology on a training set of opponent policies; (2) *Testing*: deploying the pretrained opponent model in a certain way on a testing set of opponent policies to benchmark adaptability to unknown opponents.

Different OM approaches typically have their own distinct focuses:

1. **P**retraining-**F**ocused **A**pproach (**PFA**) focuses on acquiring knowledge of responding to different opponents during pretraining and generalizing it to testing.

2. **T**esting-**F**ocused **A**pproach (**TFA**) centers on updating the pretrained opponent model during testing to reason and respond to unknown opponents effectively.

Methodologically, OM approaches based on *Representation Learning* (Jaiswal et al., 2020) and *Non-meta-gradient Meta-learning* (Duan et al., 2016) typically fall under the category of PFA:

- OM approaches based on *Representation Learning* aim to learn high-quality representations of opponent policies during pretraining to assist in policy optimization (He et al., 2016a; Hong et al., 2018; Grover et al., 2018; Papoudakis & Albrecht, 2020; Zintgraf et al., 2021; Papoudakis et al., 2021a; Papoudakis & Albrecht, 2020).

- OM approaches based on *Non-meta-gradient Meta-learning* attempt to use recurrent architectures to learn the internal structure of each opponent's policy and the differences between them during pretraining (Wang et al., 2016; Zintgraf et al., 2021).

OM approaches based on *Bayesian Learning* (Bernardo & Smith, 2009), *Meta-gradient-based Meta-learning* (Finn et al., 2017), *Shaping Opponents' Learning* (Foerster et al., 2018a), *Recursive Reasoning* (Wen et al., 2019), and *Theory of Mind* (Rabinowitz et al., 2018) methodologies typically fall under the category of TFA:

- OM approaches based on *Bayesian Learning* detect or infer the opponent policies in real-time using Bayesian methods and subsequently generate responses based on the inferred information (Bard et al., 2013; Rosman et al., 2016; Hernandez-Leal et al., 2016; Zheng et al., 2018; DiGiovanni & Tewari, 2021; Fu et al., 2022; Lv et al., 2023).

- OM approaches based on *Meta-gradient-based Meta-learning* emphasizes leveraging the well-initialized solutions obtained during pretraining in the parameter space to fine-tune and quickly adapt to the test opponents (Al-Shedivat et al., 2018; Kim et al., 2021; Wu et al., 2022b).

- OM approaches based on *Shaping Opponents' Learning* model opponents' updating gradients to estimate the mutual influence between the future opponent policy and the current self-agent's policy (Foerster et al., 2018a;b; Letcher et al., 2019; Kim et al., 2021; Lu et al., 2022; Willi et al., 2022; Zhao et al., 2022; Fung et al., 2023).

- OM approaches based on *Recursive Reasoning* simulate nested layers of beliefs, predict the opponent's behavior, and generate the best response based on the expected reasoning process of the opponent towards the self-agent (Wen et al., 2019; 2021; Dai et al., 2020; Yuan et al., 2020; Yu et al., 2022).

- OM approaches based on *Theory of Mind* concentrate on reasoning about the opponent's mental states and intentions to predict and adapt to their behavior (Von Der Osten et al., 2017; Rabinowitz et al., 2018; Raileanu et al., 2018; Yang et al., 2019).

**O**ffline **O**pponent **M**odeling (**OOM**) is a recently proposed learning paradigm that aims to use offline datasets, without interacting with the environment or opponent agents, to learn an adaptive self-agent policy that dynamically adapts to opponents based on opponent information. Jing et al. (2024a) was the first to propose OOM. It first uses representation learning methods to learn embeddings of opponent policies and then employs In-Context Learning to train an adaptive self-agent policy. Jing et al. (2024a) assumes that the offline dataset is optimal, which is a challenging assumption to satisfy

in real-world scenarios. We found that when this assumption is not met, the performance of OOM algorithms deteriorates to an unacceptable level. Our work aims to propose a general algorithmic improvement framework, TIPR, to relax this assumption and enable OOM algorithms to learn workable adaptive self-agent policies even from highly suboptimal offline datasets.

Previous online OM approaches belonging to PFA can be easily adapted to the OOM learning paradigm, as their methodologies generally do not rely on interactions with the environment or other agents. We adopt the same OOM baselines as Jing et al. (2024a), which are reasonably modified from some representative PFAs. Interestingly, Wu et al. (2024a) recently introduced Constrained Self-Play for offline policy adaptation, which shares a similar objective with OOM. However, the setup of Wu et al. (2024a) is much simpler than OOM: it assumes that interaction with the environment is still possible and that the opponent adopts a fixed, known policy during testing rather than unknown non-stationary policies.

**In-Context Learning (ICL).**     Algorithmically, **I**n-**C**ontext **L**earning (ICL) can be considered as taking a more agnostic approach by learning the learning algorithm itself (Duan et al., 2016; Wang et al., 2016; Mishra et al., 2018; Laskin et al., 2023). Recent work investigates why and how supervised pretrained Transformers perform ICL (Garg et al., 2022; Li et al., 2023b; Zhang et al., 2023; Ahn et al., 2023; Raventos et al., 2023). Xie et al. (2021) introduces a Bayesian framework explaining how ICL works. Some work proves Transformers can implement ICL algorithms via in-context gradient descent (Von Oswald et al., 2023; Akyürek et al., 2023; Bai et al., 2023).

In terms of decision-making, ICL can endow the pretrained model *Reinforcement Learning* (RL) abilities in an in-context way (Wang et al., 2016; Duan et al., 2016; Grigsby et al., 2023; Dorfman et al., 2021; Mitchell et al., 2021; Li et al., 2020; Pong et al., 2022; Laskin et al., 2023; Lee et al., 2023). Wang et al. (2016); Duan et al. (2016); Grigsby et al. (2023) focus on the online meta-RL setting with the training objective to be the total reward. Furthermore, Dorfman et al. (2021); Mitchell et al. (2021); Li et al. (2020); Pong et al. (2022) focus on offline meta-RL, and their training objectives explicitly handle the distribution shift. Laskin et al. (2023) applies autoregressive supervised learning to distill (sub-sampled) traces of a single-task RL algorithm into a task-agnostic model. Lee et al. (2023) proposes supervised pretraining to empirically and theoretically demonstrate In-Context RL abilities. Lin et al. (2024) further introduce a theoretical analysis framework to explain the principles and working conditions of In-Context RL.

Existing ICL methods in decision-making primarily focus on task adaptation in single-agent settings. Our work is among the few exploratory studies focusing on adapting to non-stationary opponents in multi-agent scenarios. Jing et al. (2024a) was the first to propose the OOM problem setting. It introduced using ICL methods to learn an adaptive self-agent policy, which conditions on opponent information to imitate self-agent policies embedded in the offline dataset. However, the suboptimality of real-world offline datasets significantly reduces the effectiveness of OOM algorithms for the opponent adaptation. To address this issue, we propose an improvement framework based on horizon-truncated in-context Q to handle the suboptimality of OOM algorithms induced by offline datasets. Inspired by the aforementioned ICL works, we train the action-value function Q in an *in-context manner*. We found that Q learned this way, compared to an unconditional Q, has higher confidence and can better handle the impact of opponent non-stationarity. Such an ICL-based design choice effectively enhances Q's ability to improve the adaptation performance of various OOM algorithms.

**Offline RL.**     The mainstream methodology in Offline RL is *Offline Conservative Learning* (OCL), which performs policy iteration with the mixed policies embedded in the offline dataset as reference policies. OCL may incorporate soft or hard conservative constraints to restrict updates within the trajectory distribution of the offline dataset. In offline single-agent RL, pessimism-driven OCL has been widely demonstrated to effectively reduce extrapolation error (Kumar et al., 2020; Fujimoto et al., 2019; Jin et al., 2021; Xiao et al., 2021; Kostrikov et al., 2021). Some have implemented OCL through statistical constraints (Wu et al., 2019; Peng et al., 2019; Brandfonbrener et al., 2021; Fujimoto & Gu, 2021; Nair et al., 2020), such as KL divergence. Other works have employed in-sample methods for conservative policy optimization (Fujimoto et al., 2019; Kostrikov et al., 2021; Ma et al., 2021; Wu et al., 2022a; Xiao et al., 2023). Additionally, some have used ensemble methods to encode the concept of conservatism (Kumar et al., 2019; Ghasemipour et al., 2021; Wu et al., 2021; Bai et al., 2021).

In this work, we propose TIPR, which improves OOM algorithms by performing IPR through a horizon-truncated in-context Q to address the suboptimality induced by offline datasets. Methodologically, TIPR is in sharp contrast to OCL, as TIPR performs policy refinement instantly during testing rather than by optimizing an objective function during offline pretraining like OCL. Additionally, compared to conservatism, TIPR places greater emphasis on policy balancing—performing policy refinement when confidence in value estimation is high while retaining the original policy when confidence is low. Notably, we found that using our proposed Truncated Q for OCL can also effectively improve the original OOM policy to some extent.

In contrast, using Original Q for OCL not only fails to enhance the original OOM policy but often completely undermines it.

**Offline Multi-Agent RL (MARL).** Offline MARL has emerged as an important research topic on multi-agent system in recent years (Cui & Du, 2022; Tian et al., 2023; Wu et al., 2023; Jiang & Lu, 2023; Formanek et al., 2023; Yang et al., 2021; Pan et al., 2022; Shao et al., 2024; Tseng et al., 2022; Meng et al., 2023; Zhu et al., 2023). This research focuses on developing effective policies solely from pre-existing datasets, without any interaction with multi-agent environments. The offline approach eliminates the need for costly or potentially hazardous online exploration, thereby enhancing the practicality of MARL for real-world applications.

An vital challenge in offline MARL is to handle the distribution shift between the offline dataset and the true transition dynamics at test time (Jiang & Lu, 2021). This challenge is also referred to as extrapolation error, where offline models may encounter out-of-distribution states or actions when deployed online. When opponents adopt policies not represented in the offline dataset, the performance of policies trained exclusively offline can deteriorate substantially. Recent studies have introduced several techniques to mitigate this challenge. Consistent with the ideas of many offline single-agent RL methods (Fujimoto et al., 2019; Kumar et al., 2020), some address extrapolation error through the concept of OCL (Yang et al., 2021; Pan et al., 2022; Shao et al., 2024). For example, Pan et al. (2022) introduced Offline MARL with Actor Rectification that uses zero-order optimization to update the actor based on the offline critic conservatively. There is also increasingly work adapting Transformer architecture for offline MARL. For instance, Tseng et al. (2022) proposed distilling knowledge from a teacher Transformer model trained with global information into decentralized student to enable effective decentralized execution. Multi-Agent Decision Transformer (Meng et al., 2023) treats MARL as a sequence modeling problem and predicts actions in an autoregressive manner. Recently, there have also been explorations of developing offline MARL frameworks based on diffusion models for centralized training with decentralized execution (Zhu et al., 2023).

Offline MARL and OOM share a common goal of deriving effective policies from offline datasets in multi-agent environments and applying them in testing scenarios to enhance efficiency and narrow the gap with real-world applications. Nevertheless, most offline MARL approaches focus on purely cooperative settings, while in OOM, we mainly focus on competitive games where the environment features opponents. Moreover, OOM, either explicitly or implicitly, must incorporate the modeling of other agents in the environment to achieve adaptability, whereas offline MARL does not place significant emphasis on this requirement. This is because OOM typically assumes a more challenging setting where the policies of other agents in the testing environment are unknown and non-stationary.

**Transformers for Decision-Making.** Growing interest has emerged in utilizing Transformers for decision-making by reframing the problem as sequence modeling. (Yang et al., 2023a; Li et al., 2023a). Chen et al. (2021) introduced *Decision Transformer* (DT), a model that predicts action sequences conditioned on returns using a causal Transformer trained on offline data. DT has opened up a series of research from a new perspective, that decision-making problems can be addressed through *Return-Conditioned Supervised Learning* (RCSL). Further research has investigated advancements such as enhanced conditioning (Furuta et al., 2022; Paster et al., 2022) and architectural improvements (Villaflor et al., 2022). Another promising avenue leverages the generality and scalability of Transformers for multi-task learning (Lee et al., 2022; Reed et al., 2022). Additionally, Transformers used in decision-making have shown potential for meta-learning (Melo, 2022).

Interestingly, recent works have introduced methods from offline single-agent RL into RCSL to address the issue of suboptimality in offline datasets (Yamagata et al., 2023; Wu et al., 2024b; Wang et al., 2024; Hu et al., 2024; Zhuang et al., 2024). The core ideas of these works can also be summarized as OCL. This work proposes TIPR, which differs from these prior works in the following aspects:

1. *Methodological Difference from OCL*: Unlike OCL, which performs iterative policy improvement during pretraining through an objective function, TIPR is the first to introduce IPR, conducting instant policy refinement during testing to improve the original policy.

2. *Not an RCSL Method*: TIPR does not belong to the RCSL class of methods, as Truncated Q is not conditioned on the Return-To-Go. Brandfonbrener et al. (2022) discussed the issues of specifying conditioning functions in RCSL methods and how unreasonable conditioning functions can lead to arbitrarily poor results.

It is worth noting that most of the previously mentioned ICL works adopt a Transformer architecture, as Transformers have a natural advantage in sequence modeling tasks (Garg et al., 2022; Li et al., 2023b; Zhang et al., 2023; Duan et al., 2016; Grigsby et al., 2023). For instance, Lee et al. (2023) proposed a Transformer-based ICL approach that outperforms

behaviors observed in the dataset, both empirically and theoretically, with respect to regret, a performance measure where DT falls short (Brandfonbrener et al., 2022; Yang et al., 2023b). Building on these insights, our Truncated Q utilizes a causal Transformer architecture to enhance the model's ICL capabilities, particularly within the domain of OM.

## B. Detailed Theoretical Analysis

### B.1. Explanations of Notations

*Table 2.* Notations in Thm. 3.1

| Notation | Description |
|---|---|
| $h$ | The random variable of truncated horizon for the **Truncated Q** function. |
| $f(h)$ | The *Empirical No Maximization Bias* (NMB) probability, *i.e.*, the probability that the action selected by the Truncated Q function aligns with the action that maximizes the true truncated expected return. |
| $\eta_0$ | A parameter used to lower bound the empirical NMB probability with the inequality $f(h) \geq 1 - 2\eta_0$, where this parameter satisfies the equation that $\Delta(h)(\sqrt{\frac{S(\ln\frac{2\|\mathbb{T}\|}{S}+1)-\ln\eta_0/4\|\mathbb{O}^1\|\|\mathbb{A}\|\|\mathcal{D}\|}{l_{\min}}} + \frac{1}{l_{\min}} + \sqrt{\frac{-\ln\eta_0/\|\mathbb{O}^1\|\|\mathbb{A}\|\|\mathcal{D}\|}{2l_{\min}}}) = \frac{U^2}{4}$. |
| $\Delta(h)$ | A term related to the Q function's fitting error over the truncated horizon; its asymptotic complexity is $O(h^2)$. |
| $S$ | The *Vapnik-Chervonenkis* (VC) dimension of the function set $\mathbb{L} := \{A(h) \leq \mathfrak{L}(o^1, a, D, G_h; \omega) \leq B(h), \omega \in \Omega\}$, where $\mathfrak{L}(o^1, a, D, G_h; \omega) := (\check{Q}_h(o^1, a, D; \omega) - G_h)^2$ denotes the loss function for learning Truncated Q function $\check{Q}_h$. $A(h)$ is the infimum of $\mathfrak{L}(o^1, a, D, G_h; \omega)$ while $B(h)$ is its supremum. $\Omega$ is the space of parameter $\omega$. Notations using sans-serif fonts all represent random variables. |
| $\|\mathbb{T}\|$ | The size (number of trajectories) of the offline dataset $\mathbb{T}$. |
| $\|\mathbb{O}^1\|$ | The cardinality of the self-agent's observation space. |
| $\|\mathbb{A}\|$ | The cardinality of the joint action space of all agents in the environment. |
| $\|\mathcal{D}\|$ | The cardinality of the In-Context Data set $\mathcal{D}$, where $\mathcal{D}$ is the set of all possible In-Context Data $D$. |
| $l_{\min}$ | Let $l(o^1, a, D)$ denote the number of samples $(o^1, a, D)$ in the offline dataset $\mathbb{T}$, then $l_{\min} = \min_{o^1, a, D} l(o^1, a, D)$ represents the minimum number of samples $(o^1, a, D)$ in the offline dataset $\mathbb{T}$. |
| $U$ | A constant satisfies equation that $\min_{o^1, a^{-1}, D}(G_h(o^1, a^{1*}, a^{-1}, D) - G_h(o^1, a^{1+}, a^{-1}, D)) = U$, where $a^{1*}$ is the self-agent action that maximizes $G_h$, and $a^{1+}$ is the self-agent action with the second-highest $G_h$. |

### B.2. Proof of Thm. 3.1

**Theorem 3.1.** *For any given $o^1, a^{-1}, D$, we have $f(h) \geq \mathfrak{f}(h) := 1 - 2\eta_0$, where $\eta_0$ satisfies that*

$$\Delta(h)(\sqrt{\frac{S(\ln\frac{2\|\mathbb{T}\|}{S}+1) - \ln\eta_0/4\|\mathbb{O}^1\|\|\mathbb{A}\|\|\mathcal{D}\|}{l_{min}}} + \frac{1}{l_{min}} + \sqrt{\frac{-\ln\eta_0/\|\mathbb{O}^1\|\|\mathbb{A}\|\|\mathcal{D}\|}{2l_{min}}}) = \frac{U^2}{4}. \tag{7}$$

*The asymptotic complexity of the term $\Delta(h)$ is $O(h^2)$.*

*Proof.* First, we define some notation for convenience in the derivation. Let the truncated horizon random variable be denoted as h, and let the neural network for Truncated Q be represented as $\check{Q}_h$, with parameters $\omega$ and parameter space $\Omega$.

Let the random variable for the self-agent's observation be denoted as $o^1$, the random variable for the joint actions of all agents as $a$, and the random variable for the In-Context Data as $D$. Let the set of all possible In-Context Data denoted as $\mathcal{D} := \{D = (o_m^{-1}, a_m^{-1})_{m=1}^M | (o^{-1}, a^{-1}) \in (\mathbb{O}^{-1} \times \mathbb{A}^{-1})\}$.

Without loss of generality, we use *Mean Squared Error* (MSE) to define the loss function for learning $\breve{Q}_h$ as follows:

$$\mathfrak{L}(o^1, a, D, G_h; \omega) := (\breve{Q}_h(o^1, a, D; \omega) - G_h)^2. \tag{8}$$

Assuming that the infimum of Eq. (8) is $A(h)$ and the supremum of Eq. (8) is $B(h)$, we define the difference between the supremum and supremum as $\Delta(h) := B(h) - A(h)$. Under a general assumption, let the environment reward random variable $r$ follow a certain distribution with values in $[a, b]$. Then, we have $ah \leq G_h \leq bh$. It is straightforward to derive that the asymptotic complexity of $\Delta(h)$ is $O(h^2)$.

Ideally, we aim to find the optimal parameter $\omega^*$ that minimizes the following *risk functional* $\mathcal{R}(\omega)$ within the function set $\mathbb{L} := \{A(h) \leq \mathfrak{L}(o^1, a, D, G_h; \omega) \leq B(h), \omega \in \Omega\}$.

$$\mathcal{R}(\omega) = \sum_{o^1, a, D} P(o^1, a, D) \int_{G_h} \mathfrak{L}(o^1, a, D, G_h; \omega) dF(G_h | o^1, a, D) = \sum_{o^1, a, D} P(o^1, a, D) \mathcal{R}(o^1, a, D | \omega). \tag{9}$$

Here, $F$ represents the cumulative distribution function, and $\mathcal{R}(o^1, a, D | \omega)$ denotes the *per-sample risk functional*.

However, since the true distribution of $(o^1, a, D)$ is unknown, we can only apply the **Empirical Risk Minimization (ERM)** principle on the pre-collected offline dataset $\mathbb{T}$ to find the most suitable $\omega$. That is, we seek the parameter $\omega^+$ that minimizes the following *empirical risk functional* $\mathcal{R}_{emp}(\omega)$.

$$\mathcal{R}_{emp}(\omega) = \frac{1}{|\mathbb{T}|} \sum_{i=1}^{|\mathbb{T}|} \mathfrak{L}(o_{(i)}^1, a_{(i)}, D_{(i)}, G_{h(i)}; \omega) = \frac{1}{|\mathbb{T}|} \sum_{o^1, a, D} l(o^1, a, D) \mathcal{R}_{emp}(o^1, a, D | \omega). \tag{10}$$

Here, $l(o^1, a, D)$ represents the number of samples $(o^1, a, D)$ in the offline dataset $\mathbb{T}$, and $\mathcal{R}_{emp}(o^1, a, D | \omega)$ denotes the *per-sample empirical risk functional*.

Next, we introduce two lemmas to assist in the proof.

**Lemma B.1.** *(Vapnik, 1998) Assume that $\{A \leq \mathfrak{L}(x; \omega) \leq B, \omega \in \Omega\}$ is a measurable, bounded real function set whose indicator set satisfies the measurability condition. Let $\Phi = \{\phi | \inf_{\omega, x} \mathfrak{L}(x; \omega) \leq \phi \leq \sup_{\omega, x} \mathfrak{L}(x; \omega)\}$, and denote the annealed entropy of the indicator set as $\mathcal{H}_{ann}^{\Omega, \Phi}$. Then, the following inequality holds:*

$$P\{\sup_{\omega \in \Omega}(\int_x \mathfrak{L}(x; \omega) dF(x) - \frac{1}{|\mathbb{T}|} \sum_{i=1}^{|\mathbb{T}|} \mathfrak{L}(x_{(i)}; \omega)) > \epsilon\} \leq 4 \exp\{(\frac{\mathcal{H}_{ann}^{\Omega, \Phi}(2|\mathbb{T}|)}{|\mathbb{T}|} - \frac{(\epsilon - 1/|\mathbb{T}|)^2}{(B - A)^2})|\mathbb{T}|\}. \tag{11}$$

*Eq. (11) can be equivalently expressed as follows.*

*For $\forall \omega$, the following inequality holds with probability at least $1 - \eta$:*

$$\mathcal{R}(\omega) \leq \mathcal{R}_{emp}(\omega) + (B - A) \sqrt{\frac{\mathcal{H}_{ann}^{\Omega, \Phi}(2|\mathbb{T}|) - \ln \eta/4}{|\mathbb{T}|} + \frac{1}{|\mathbb{T}|}}. \tag{12}$$

**Lemma B.2.** *(Vapnik, 1998) Let the Vapnik-Chervonenkis (VC) dimension of the function set $\{A \leq \mathfrak{L}(x; \omega) \leq B, \omega \in \Omega\}$ in Thm. B.1 be denoted as $S$. Then, the following inequality holds:*

$$\mathcal{H}_{ann}^{\Omega, \Phi}(|\mathbb{T}|) \leq S(\ln \frac{|\mathbb{T}|}{S} + 1). \tag{13}$$

Substituting the previously defined loss function Eq. (8) and sample data $(o^1, a, D)$ into Thm. B.1 and Thm. B.2, we have:

$$P\{\sup_{\omega \in \Omega} \max_{o^1, a, D}(\int_{G_h} \mathfrak{L}(o^1, a, D, G_h; \omega) dF(G_h | o^1, a, D) - \frac{1}{l(o^1, a, D)} \sum_{i=1}^{l(o^1, a, D)} \mathfrak{L}(o_{(i)}^1, a_{(i)}, D_{(i)}, G_{h(i)}; \omega)) > \epsilon\}$$

$$\le \sum_{\mathsf{o}^1,\mathsf{a},\mathsf{D}} P\{\sup_{\omega\in\Omega}(\int_{\mathsf{G_h}}\mathfrak{L}(\mathsf{o}^1,\mathsf{a},\mathsf{D},\mathsf{G_h};\omega)\mathrm{d}F(\mathsf{G_h}|\mathsf{o}^1,\mathsf{a},\mathsf{D}) - \frac{1}{l(\mathsf{o}^1,\mathsf{a},\mathsf{D})}\sum_{i=1}^{l(\mathsf{o}^1,\mathsf{a},\mathsf{D})}\mathfrak{L}(\mathsf{o}^1_{(i)},\mathsf{a}_{(i)},\mathsf{D}_{(i)},\mathsf{G}_{\mathsf{h}(i)};\omega)) > \epsilon\} \quad \text{(14a)}$$

$$\le \sum_{\mathsf{o}^1,\mathsf{a},\mathsf{D}} 4\exp\{(\frac{\mathcal{H}_{\mathrm{ann}}^{\Omega,\Phi}(2l(\mathsf{o}^1,\mathsf{a},\mathsf{D}))}{l(\mathsf{o}^1,\mathsf{a},\mathsf{D})} - \frac{(\epsilon - 1/l(\mathsf{o}^1,\mathsf{a},\mathsf{D}))^2}{(\Delta(\mathsf{h}))^2})l(\mathsf{o}^1,\mathsf{a},\mathsf{D})\} \quad \text{(14b)}$$

$$\le 4|\mathbb{O}^1||\mathbb{A}||\mathcal{D}|\exp\{S(\ln\frac{2|\mathbb{T}|}{S}+1) - \frac{(\epsilon-1/l_{\min})^2}{(\Delta(\mathsf{h}))^2}l_{\min}\}. \quad \text{(14c)}$$

Here, $l_{\min} = \min_{\mathsf{o}^1,\mathsf{a},\mathsf{D}} l(\mathsf{o}^1,\mathsf{a},\mathsf{D})$ represents the minimum number of samples $(\mathsf{o}^1,\mathsf{a},\mathsf{D})$ in the offline dataset $\mathbb{T}$. Eq. (14a) clearly holds. Eq. (14b) can be derived from Eq. (11) in Thm. B.1. Eq. (14c) follows from Eq. (13) in Thm. B.2, where $S$ represents the VC dimension of $\mathbb{L}$.

According to Eq. (12) in Thm. B.2, the above Eq. (14) can be equivalently written as follow.

For $\forall\omega,\mathsf{o}^1,\mathsf{a},\mathsf{D}$, the following inequality holds with probability at least $1-\eta$:

$$\mathcal{R}(\mathsf{o}^1,\mathsf{a},\mathsf{D}|\omega) - \mathcal{R}_{\mathrm{emp}}(\mathsf{o}^1,\mathsf{a},\mathsf{D}|\omega) \le \Delta(\mathsf{h})\sqrt{\frac{S(\ln\frac{2|\mathbb{T}|}{S}+1)-\ln\eta/4|\mathbb{O}^1||\mathbb{A}||\mathcal{D}|}{l_{\min}} + \frac{1}{l_{\min}}}. \quad \text{(15)}$$

According to **Hoeffding's Inequality** (Hoeffding, 1994), we have:

$$P\{\max_{\mathsf{o}^1,\mathsf{a},\mathsf{D}}(\int_{\mathsf{G_h}}\mathfrak{L}(\mathsf{o}^1,\mathsf{a},\mathsf{D},\mathsf{G_h};\omega)\mathrm{d}F(\mathsf{G_h}|\mathsf{o}^1,\mathsf{a},\mathsf{D}) - \frac{1}{l(\mathsf{o}^1,\mathsf{a},\mathsf{D})}\sum_{i=1}^{l(\mathsf{o}^1,\mathsf{a},\mathsf{D})}\mathfrak{L}(\mathsf{o}^1_{(i)},\mathsf{a}_{(i)},\mathsf{D}_{(i)},\mathsf{G}_{\mathsf{h}(i)};\omega)) > \epsilon\}$$

$$\le \sum_{\mathsf{o}^1,\mathsf{a},\mathsf{D}} P\{\int_{\mathsf{G_h}}\mathfrak{L}(\mathsf{o}^1,\mathsf{a},\mathsf{D},\mathsf{G_h};\omega)\mathrm{d}F(\mathsf{G_h}|\mathsf{o}^1,\mathsf{a},\mathsf{D}) - \frac{1}{l(\mathsf{o}^1,\mathsf{a},\mathsf{D})}\sum_{i=1}^{l(\mathsf{o}^1,\mathsf{a},\mathsf{D})}\mathfrak{L}(\mathsf{o}^1_{(i)},\mathsf{a}_{(i)},\mathsf{D}_{(i)},\mathsf{G}_{\mathsf{h}(i)};\omega) > \epsilon\} \quad \text{(16a)}$$

$$\le \sum_{\mathsf{o}^1,\mathsf{a},\mathsf{D}} \exp\{-\frac{2\epsilon^2 l(\mathsf{o}^1,\mathsf{a},\mathsf{D})}{(\Delta(\mathsf{h}))^2}\} \quad \text{(16b)}$$

$$\le |\mathbb{O}^1||\mathbb{A}||\mathcal{D}|\exp\{-\frac{2\epsilon^2 l_{\min}}{(\Delta(\mathsf{h}))^2}\}. \quad \text{(16c)}$$

Eq. (16a) and Eq. (16c) clearly hold. The above Eq. (16) can be equivalently written as follow.

For $\forall\mathsf{o}^1,\mathsf{a},\mathsf{D}$, the following inequality holds with probability at least $1-\eta$:

$$\mathcal{R}_{\mathrm{emp}}(\mathsf{o}^1,\mathsf{a},\mathsf{D}|\omega^*) - \mathcal{R}(\mathsf{o}^1,\mathsf{a},\mathsf{D}|\omega^*) \le \Delta(\mathsf{h})\sqrt{\frac{-\ln\eta/|\mathbb{O}^1||\mathbb{A}||\mathcal{D}|}{2l_{\min}}}. \quad \text{(17)}$$

As mentioned earlier, in practice, we can only find $\omega^+$ by minimizing $\mathcal{R}_{\mathrm{emp}}(\omega)$ (see Eq. (10)). However, strictly speaking, we should find $\omega^+$ by optimizing the following objective:

$$\min_\omega \max_{\mathsf{o}^1,\mathsf{a},\mathsf{D}}(\mathcal{R}_{\mathrm{emp}}(\mathsf{o}^1,\mathsf{a},\mathsf{D}|\omega) - \mathcal{R}_{\mathrm{emp}}(\mathsf{o}^1,\mathsf{a},\mathsf{D}|\omega^*)). \quad \text{(18)}$$

Without loss of generality, assuming that $\omega^+$ is found using Eq. (18), we can derive that:

$$\max_{\mathsf{o}^1,\mathsf{a},\mathsf{D}}(\mathcal{R}_{\mathrm{emp}}(\mathsf{o}^1,\mathsf{a},\mathsf{D}|\omega^+) - \mathcal{R}_{\mathrm{emp}}(\mathsf{o}^1,\mathsf{a},\mathsf{D}|\omega^*)) \le \max_{\mathsf{o}^1,\mathsf{a},\mathsf{D}}(\mathcal{R}_{\mathrm{emp}}(\mathsf{o}^1,\mathsf{a},\mathsf{D}|\omega^*) - \mathcal{R}_{\mathrm{emp}}(\mathsf{o}^1,\mathsf{a},\mathsf{D}|\omega^*)) = 0. \quad \text{(19)}$$

Based on Eqs. (15), (17) and (19), we can derive the following formula.

The following inequality holds with probability at least $1-\eta$:

$$\max_{\mathsf{o}^1,\mathsf{a},\mathsf{D}}(\mathcal{R}(\mathsf{o}^1,\mathsf{a},\mathsf{D}|\omega^+) - \mathcal{R}(\mathsf{o}^1,\mathsf{a},\mathsf{D}|\omega^*))$$

$$\leq \max_{o^1,a,D}(\mathcal{R}(o^1,a,D|\omega^+) - \mathcal{R}_{\text{emp}}(o^1,a,D|\omega^+))$$

$$+ \max_{o^1,a,D}(\mathcal{R}_{\text{emp}}(o^1,a,D|\omega^+) - \mathcal{R}_{\text{emp}}(o^1,a,D|\omega^*))$$

$$+ \max_{o^1,a,D}(\mathcal{R}_{\text{emp}}(o^1,a,D|\omega^*) - \mathcal{R}(o^1,a,D|\omega^*)) \tag{20a}$$

$$\leq \Delta(h)(\sqrt{\frac{S(\ln\frac{2|\mathbb{T}|}{S}+1) - \ln\eta/4|\mathbb{O}^1||\mathbb{A}||\mathcal{D}|}{l_{\min}} + \frac{1}{l_{\min}}} + \sqrt{\frac{-\ln\eta/|\mathbb{O}^1||\mathbb{A}||\mathcal{D}|}{2l_{\min}}})$$

$$+ \max_{o^1,a,D}(\mathcal{R}_{\text{emp}}(o^1,a,D|\omega^+) - \mathcal{R}_{\text{emp}}(o^1,a,D|\omega^*)) \tag{20b}$$

$$\leq \delta := \Delta(h)(\sqrt{\frac{S(\ln\frac{2|\mathbb{T}|}{S}+1) - \ln\eta/4|\mathbb{O}^1||\mathbb{A}||\mathcal{D}|}{l_{\min}} + \frac{1}{l_{\min}}} + \sqrt{\frac{-\ln\eta/|\mathbb{O}^1||\mathbb{A}||\mathcal{D}|}{2l_{\min}}}). \tag{20c}$$

Eq. (20a) clearly holds. Eq. (20b) can be derived from Eq. (15) and Eq. (17). Eq. (20c) follows from Eq. (19).

From Eq. (20), we have:

$$\forall o^1,a,D, \quad \mathcal{R}(o^1,a,D|\omega^+) - \mathcal{R}(o^1,a,D|\omega^*) \leq \delta. \tag{21}$$

According to the **Bias-Variance Decomposition** (James et al., 2013), we have:

$$\forall o^1,a,D, \quad \mathcal{R}(o^1,a,D|\omega) = \mathbb{E}_{G_h}\mathfrak{L}(o^1,a,D,G_h;\omega) = (\breve{Q}_h(o^1,a,D;\omega) - \mathbb{E}G_h)^2 + \mathbb{V}G_h. \tag{22}$$

Substituting Eq. (22) into Eq. (21), we obtain:

$$\forall o^1,a,D, \quad (\breve{Q}_h(o^1,a,D;\omega^+) - \mathbb{E}G_h)^2 - (\breve{Q}_h(o^1,a,D;\omega^*) - \mathbb{E}G_h)^2 \leq \delta. \tag{23}$$

When the neural network $\breve{Q}_h$ converges, the bias of $\omega^*$, *i.e.*, the second term of the left side in Eq. (23), can be considered $0$. Therefore, we can derive the following formula.

For $\forall o^1,a,D$, the following inequality holds with probability at least $1 - \eta$:

$$|\breve{Q}_h(o^1,a,D;\omega^+) - \mathbb{E}G_h| \leq \sqrt{\delta}. \tag{24}$$

Assume that

$$\min_{o^1,a^{-1},D}(G_h(o^1,a^{1*},a^{-1},D) - G_h(o^1,a^{1+},a^{-1},D)) = U, \tag{25}$$

where $a^{1*}$ is the self-agent action that maximizes $G_h$, and $a^{1+}$ is the self-agent action with the second-highest $G_h$. Observing Eqs. (24) and (25), it is straightforward to derive the following formula.

When $2\sqrt{\delta} = U$, there is at least a probability of $1 - 2\eta_0$ that $\breve{Q}_h$ selects the action with the highest $\mathbb{E}G_h$, *i.e.*,

$$f(h) = P(\arg\max_{a^1}\breve{Q}_h = \arg\max_{a^1}\mathbb{E}G_h) \geq \mathfrak{f}(h) := 1 - 2\eta_0. \tag{26}$$

Within, $\eta_0$ satisfies that:

$$\Delta(h)(\sqrt{\frac{S(\ln\frac{2|\mathbb{T}|}{S}+1) - \ln\eta_0/4|\mathbb{O}^1||\mathbb{A}||\mathcal{D}|}{l_{\min}} + \frac{1}{l_{\min}}} + \sqrt{\frac{-\ln\eta_0/|\mathbb{O}^1||\mathbb{A}||\mathcal{D}|}{2l_{\min}}}) = \frac{U^2}{4}. \tag{27}$$

This concludes the proof. $\qquad\square$

## B.3. Proof and Intuitive Analysis of Prop. 3.2

**Proposition 3.2.** *For any given $o^1, a^{-1}, D$, there exists an optimal truncated horizon $h^* \in [T]$ that maximizes $\mathfrak{f}(h)g(h)$, i.e., the lower bound of $y(h)$. Specifically, there exist functions $g$ such that $h^* \neq T$.*

*Proof.* To ensure that Truncated Q selects the action with the highest $\mathbb{E}G_T$ with maximum probability during policy refinement, we need to find the optimal $h \in [T]$ that maximizes the *Overall No Maximization Bias (NMB) Probability* $y(h)$.

However, directly maximizing $y(h)$ is intractable, and we need to perform further transformations. Based on Eq. (5), we can maximize its lower bound $f(h)g(h)$. For any given $o^1, a^{-1}, D$, this objective can be written as:

$$\max_h f(h)g(h). \tag{28}$$

Here, the *Empirical Risk NMB Probability* $f(h)$ is determined by the neural network's fitting process on the dataset $\mathbb{T}$, while the *Natural NMB Probability* $g(h)$ is dictated by the intrinsic properties of the environment's reward structure.

Eq. (28) is still intractable, requiring further transformation. According to Thm. 3.1, for any given $o^1, a^{-1}, D$, $f(h)$ has a tight lower bound denoted as $\mathfrak{f}(h) = 1 - 2\eta_0$. The variation of $\mathfrak{f}(h)$ with respect to $h$ can approximately characterize the variation of $f(h)$. Thus, we can optimize the following objective as an approximation to optimize Eq. (28):

$$\max_h \mathfrak{f}(h)g(h). \tag{29}$$

As $h$ increases, $\Delta(h)$ also monotonically increases, leading to a monotonic increase in $\eta_0$, which in turn causes $\mathfrak{f}(h)$ to monotonically decrease. Therefore, $\mathfrak{f}(h)$ is a strictly monotonically decreasing function with respect to $h$.

Since $g(h)$ is determined by the environment, we cannot explicitly formulate it. However, upon observation, it is easy to conclude that $g(h)$ satisfies certain properties:

- When $h = 1$, $g(h)$ attains its minimum value, denoted as $g_{\min}$.

- When $h = T$, $g(h)$ attains its maximum value of 1.

- When $1 < h < T$, we have $g_{\min} \leq g(h) \leq 1$.

It can be observed that as $h$ increases, the function $g(h)$ generally exhibits an upward trend. Based on these properties, it can be derived that there exists an $h^* \in [T]$ that satisfies Eq. (29).

Specifically, there exist functions $g$ such that $h^* \neq T$. Below, we provide some examples:

- When $g(h) = 1$, it clearly holds that $h^* = 1$.

- There exist strictly monotonically increasing functions $g(h)$ such that $g_{\min} < g(h^*) < 1$ and $1 < h^* < T$.

This concludes the proof. $\square$

**Intuitive Analysis.** For better intuitive understanding, we further transform Eq. (29) for analysis. It is known that for any given $o^1, a^{-1}, D$, the following holds:

$$y(h) \geq \mathfrak{f}(h)g(h). \tag{30}$$

Therefore, it is straightforward to derive that:

$$-\ln y(h) \leq (-\ln \mathfrak{f}(h)) + (-\ln g(h)). \tag{31}$$

Maximizing $y(h)$ can be achieved by optimizing the following objective:

$$\min_h (-\ln \mathfrak{f}(h)) + (-\ln g(h)). \tag{32}$$

Fig. 7 presents a schematic illustration of the variation in NMB probability as $h$ changes when $g$ is a certain strictly monotonically increasing function. In the case shown in Fig. 7, the optimal $h$ needs to balance the trade-off between $-\ln \mathfrak{f}(h)$ and $-\ln g(h)$, thereby approximately minimizing $-\ln y(h)$, which in turn maximizes $y(h)$. It can be intuitively observed that the optimal value of $h$ is not necessarily always $T$.

With the aid of Fig. 7, we can further intuitively analyze the benefits of IPR making refinement decisions based on value confidence $\breve{Q}_C$, rather than unthinkingly refining the original policy in all cases. In the practical implementation of Truncated

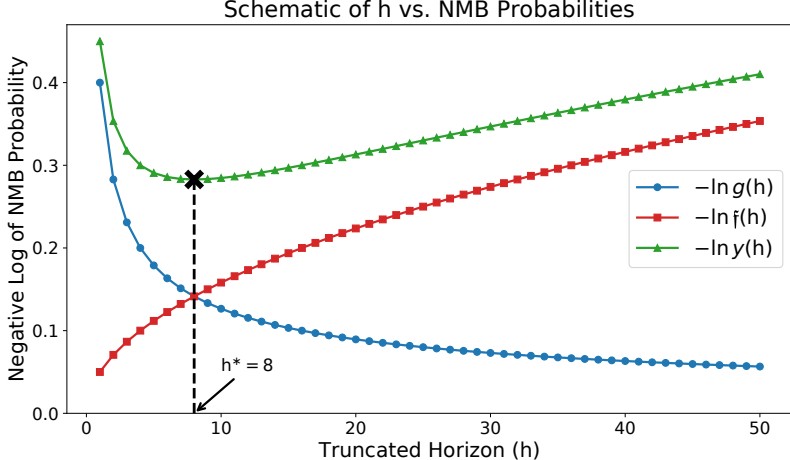

*Figure 7.* Illustration of the variation in NMB probability as h changes when $g$ is a certain strictly monotonically increasing function.

Q, we empirically set a truncated horizon $H$. However, the true h$^*$ could potentially be a much larger value. In this scenario, the $y(H)$ values for many given samples $(o^1, a^{-1}, D)$ may be very small, resulting in a low probability of selecting the action with the highest $\mathbb{E}G_T$ after policy refinement. At this point, rather than using Truncated Q for refinement, it would be better to retain the original policy. Building on this reasoning, IPR uses value confidence $\check{Q}_C$ to balance the decision of whether to perform policy refinement.

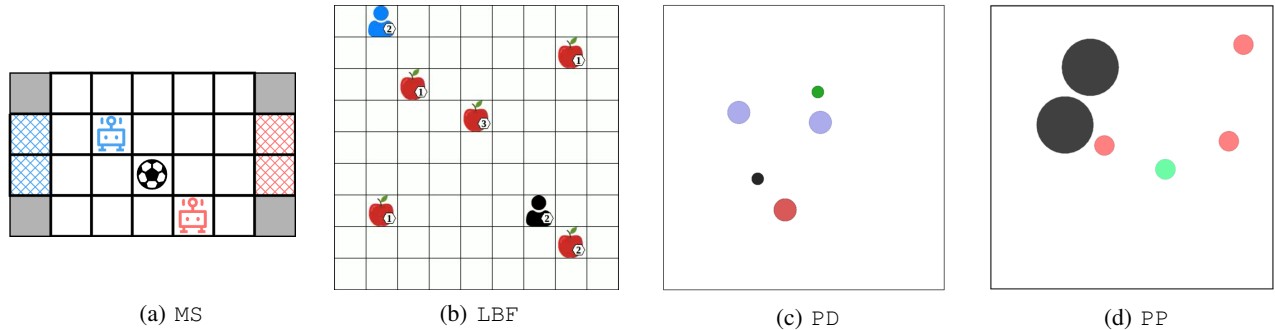

(a) MS        (b) LBF        (c) PD        (d) PP

*Figure 8.* The environmental benchmarks.

## C. Detailed Introductions of the Environments

**Markov Soccer (MS)** MS (Littman, 1994; Lanctot et al., 2019) is a sparse-reward two-player zero-sum game with a discrete state space. The game is played on a $4 \times 7$ grid in Fig. 8(a). The game begins with the self-agent (blue) and the opponent (red) in random squares in the left and right half (except the goals), respectively, and the ball goes to one of them randomly or none of them to a random square in the middle column. The players have five actions: move N, S, W, E, or stand still (*i.e.*, NO-OP). An action is considered invalid if it leads the player to a shaded square (grey) or outside the border. If a player has the ball, possession will be exchanged and transition will not take place when the two players move into the same square. Both the self-agent's and the opponent's objectives are taking the ball into the *other's goal* (the self-agent's goal is blue, and the opponent's goal is red), and the game ends when this happens. The reward is sparse, providing $+10$ for success, $-10$ for failure, and $0$ for instances where the maximum timestep limit is exceeded, only at the end of the games. For specific implementation of this environment, we adopt the open-source code of *OpenSpiel*, which is available at https://github.com/deepmind/open_spiel.

**Level-Based Foraging (LBF)** LBF (Christianos et al., 2020; Papoudakis et al., 2021b) is a sparse-reward two-player mixed-incentive game with a discrete state space, as shown in Fig. 8(b). The game is played on a $9 \times 9$ grid with the self-agent (in blue) and the opponent (in black), along with five apples (in red). At the beginning of each episode, the two players and the five apples are randomly generated in the environment and assigned a level marked in their bottom-right corner. The goal of the self-agent is to eat as many apples as possible. All players can move in four directions or eat an apple. Eating an apple can be successfully done only under the following conditions: one or two players are around the apple, and all players who take the action of eating an apple have a summed level at least equal to the level of the apple. The environment has sparse rewards, representing the players' contributions to eating all the apples in the environment. The environment is essentially a long-term social dilemma and can be viewed as an extension of the Prisoner's Dilemma (Axelrod, 1980). The challenge in this environment is that the self-agent must learn to cooperate to eat high-level apples while greedily eating low-level apples simultaneously. For specific implementation of this environment, we adopt the open-source code of *lb-foraging*, which is available at https://github.com/semitable/lb-foraging.

**Physical Deception (PD)** PD (Lowe et al., 2017) is a sparse-reward three-player non-zero-sum game with a continuous state space. PD includes the self-agent (red), two opponents (purple), one normal landmark (black), and one target landmark (green) shown in Fig. 8(c). All agents observe position of landmarks and other agents. There are no borders present on the entire game map. The self-agent's objective is to hit the target landmark which is *unknown* to him (*i.e.*, the self-agent does not know which one of the landmarks is the target one) as many times as possible. The opponents' objective is to minimize the times that the self-agent hits the target landmark while maximizing the times that they themselves hit the target landmark. Therefore, the opponents need to cleverly deceive the self-agent, while the self-agent must astutely detect the opponents' deception. When the self-agent hits the target landmark, it receives a sparse reward of $10$, while the opponents receive $-10$. When the opponents hit the target landmark, they receive a sparse reward of $10$. For specific implementation of this environment, we adopt the open-source code of *Multi-Agent Particle Environment*, which is available at https://github.com/openai/multiagent-particle-envs. Additionally, we introduce a collision mechanism to the original environment to increase the randomness and dynamics of the environment.

**Predator Prey (PP)** PP (Lowe et al., 2017) is a sparse-reward four-player non-zero-sum game with a continuous state space, as shown in Fig. 8(d). The environment consists of three predators (in red), one prey (in green), and two obstacles (in black). The goal of the predators is to capture (*i.e.*, collide with) the prey as much as possible, while the goal of the prey is to be captured as little as possible. The environment features sparse rewards, where each time a predator captures the prey, the capturing predator receives a reward of $10$, and the prey receives a reward of $-10$. Additionally, the environment provides a very small, dense reward to the prey to prevent it from running out of the map boundaries. Here, the prey is the self-agent, and the three predators serve as the opponents. From the perspective of the self-agent, the environment is highly unstable, as there are three opponents with unknown policies in the environment. The challenge in this environment is that the self-agent must model the behavior of three opponents simultaneously and adapt to various potential coordination strategies employed by the opponents (*e.g.*, surrounding from three different directions). For specific implementation of this environment, we adopt the open-source code of *Multi-Agent Particle Environment*, which is available at https://github.com/openai/multiagent-particle-envs.

# D. Diversity of Opponent Policies

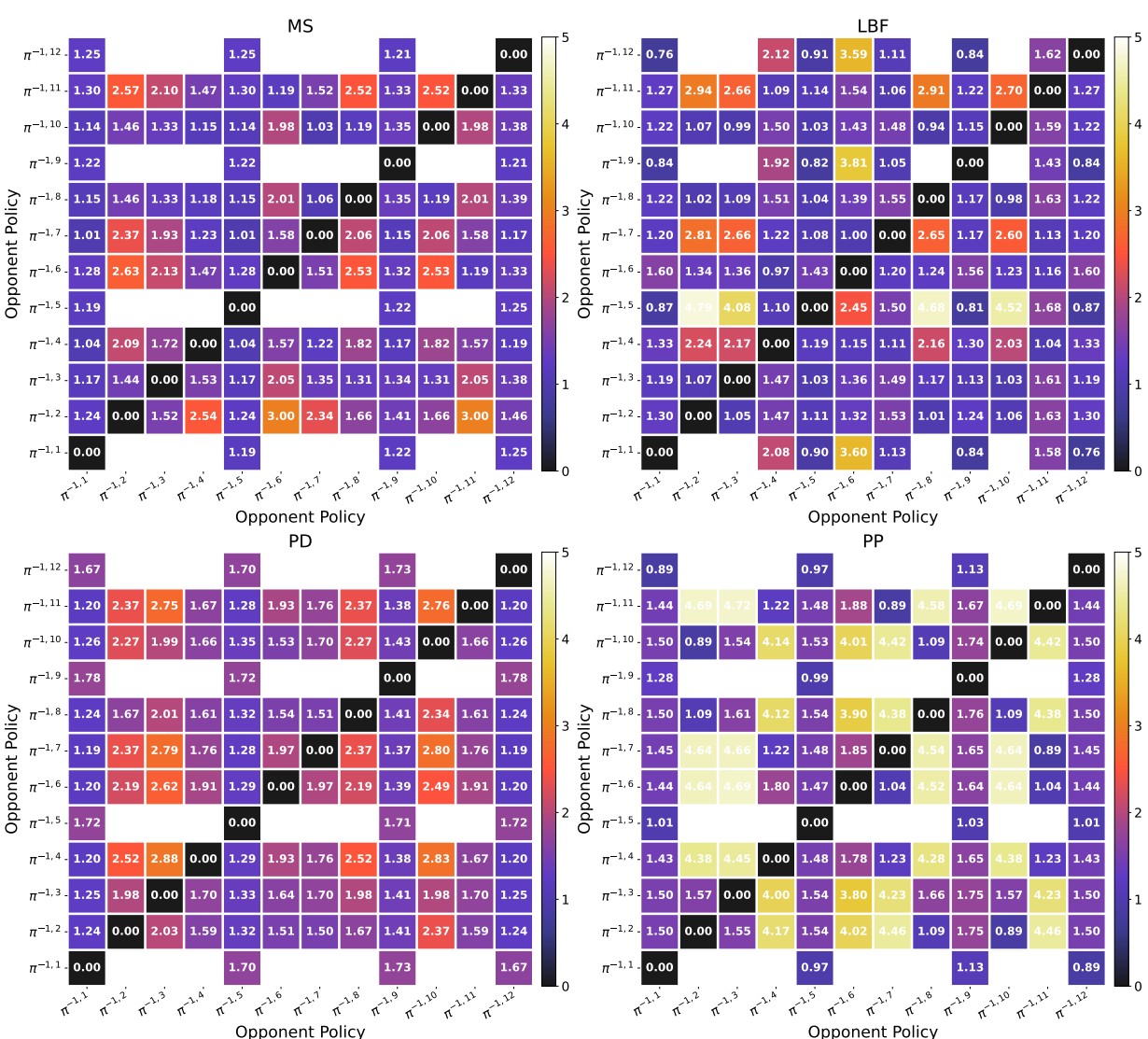

*Figure 9.* Pair-wise expected KL divergence of all the opponent policies used to construct the offline datasets $\mathbb{T}$.

As mentioned in Sec. 4.1, we run the Maximum Entropy Population-based training algorithm (MEP) (Zhao et al., 2023) to

generate a diversified opponent policy population. Nevertheless, a quantitative analysis is still necessary to measure the similarity/dissimilarity between different opponent policies used to construct the offline datasets $\mathbb{T}$.

From the MEP population, 12 policies are selected as pretraining opponents (Seen Set), and 8 policies are selected as testing opponents in the Unseen Set. We calculate the pair-wise expected KL divergence between different pretraining opponents to measure their dissimilarity. The results for MS, LBF, PD, and PP are shown in Fig. 9, respectively.

For any given policies $\pi^{-1,i}$ and $\pi^{-1,j} \in$ Seen Set, we estimate the expected KL divergence between them by:

$$D_{KL}(\pi^{-1,i}||\pi^{-1,j}) = \mathbb{E}_{o^{-1} \sim P(o^{-1})} \left[ \sum_{a^{-1} \in \mathbb{A}^{-1}} \pi^{-1,i}(a^{-1}|o^{-1}) \cdot \log \frac{\pi^{-1,i}(a^{-1}|o^{-1})}{\pi^{-1,j}(a^{-1}|o^{-1})} \right]. \tag{33}$$

Here, $P(o^{-1})$ denotes the opponent observation distribution. Ideally, $P(o^{-1})$ should cover the entire opponent observation space $\mathbb{O}^{-1}$. However, in practical situations, covering the entire opponent observation space $\mathbb{O}^{-1}$ in even slightly large environments can be intractable.

To maximize the coverage of the opponent observation space by $P(o^{-1})$, we employ the following sampling method: Within the Seen Set, there are a total of 12 opponent policies. For each opponent policy $\pi^{-1,k}$, we sample 1200 episodes. In these 1200 episodes, the opponents' policy are fixed to $\pi^{-1,k}$ while the self-agent traverses through all the opponent policies, resulting in the self-agent using per opponent policy for 100 episodes.

In Fig. 9, the lighter the color in the heatmap, the higher the KL divergence value, indicating a lower similarity between the two policies. Assuming a dissimilarity threshold of $1.0$ (*i.e.*, two policies are dissimilar if their expected KL divergence is greater than $1.0$), over $80\%$ of the expected KL divergence values in all environments exceed this threshold. This indicates that the pretraining opponent policies are generally well-distinguished from one another. The dissimilarity rates for MS, LBF, PD, and PP are $92.35\%$, $80.56\%$, $91.67\%$, and $85.42\%$, respectively. Overall, we can consider the MEP opponent policies we select to construct the offline datasets $\mathbb{T}$ to be adequately diverse.

## E. Neural Architecture Designs

### E.1. Neural Architectures of Truncated Q

For the horizon-truncated in-context action-value function we proposed, *i.e.*, **Truncated Q**, we adopt the neural architecture design as follows:

- *Backbone*: The backbone of the Truncated Q is mainly implemented based on the causal Transformer, *i.e.*, GPT2 (Radford et al., 2019) model of Hugging Face (Wolf et al., 2020). Both the Encoder and Decoder of the Truncated Q's neural architecture adopt this backbone. The backbone is a ***GPT2 model*** composed of 3 self-attention blocks. Each self-attention block consists of a single-head attention layer and a feed-forward layer. The Decoder's self-attention block includes an additional single-head cross-attention layer. Residual connections (He et al., 2016b) and LayerNorm (Ba et al., 2016) are utilized after each layer in the self-attention block. Within each attention layer, dropout (Srivastava et al., 2014) is added to the residual connection and attention weight. In the backbone, the feed-forward layer consists of a fully connected layer that increases the number of hidden layer nodes and a projection layer that recovers the number of hidden layer nodes. Except for the fully connected layer in the feed-forward layer, which consists of 128 nodes with GELU (Hendrycks & Gimpel, 2016) activation functions, the other hidden layers are composed of 32 nodes without activation functions.

- *Encoder*: The opponents' observations $o^{-1}$ and actions $a^{-1}$ are fed into modality-specific linear layers, and a positional episodic timestep encoding is added. We adopt the same timestep encoding as in (Chen et al., 2021). Then, we use a *fusion linear layer* to fuse the $o^{-1}, a^{-1}$ embedding tokens at each timestep into *fused embedding tokens*. Finally, the sequences of fused embedding tokens are fed into the backbone, which autoregressively outputs the *per-timestep embedding tokens* $z^{-1}$ corresponding to each $(o^{-1}, a^{-1})$ tuple using a causal self-attention mask. The per-timestep embedding tokens $z^{-1}$ are inputed as ***key*** and ***value*** into the cross-attention layers of the Decoder. The fusion linear layer comprise 32 nodes without activation functions. The modality-specific linear layers for the opponents' observations $o^{-1}$ and actions $a^{-1}$ comprise 32 nodes with ELU (Clevert et al., 2015) activation functions.

- *Decoder*: Self-agent observations $o^1$, opponents actions $a^{-1}$, self-agent actions $a^1$, and self-agent rewards $r^1$ are fed into modality-specific linear layers. A positional episodic timestep encoding is added. We adopt the same timestep

encoding as in Chen et al. (2021). In addition, an agent index encoding is added to each token to distinguish the inputs from different agents. The sequences of embedded tokens are fed into the cross-attention layers of the backbone as **query**, which autoregressively outputs the action-value predictions $\check{Q}$ at the positions of self-agent actions $a^1$ tokens using a causal self-attention mask. The Opponent Imitator $\pi_{\text{OI}}$ of the Decoder autoregressively predicts opponents actions $a^{-1}$ at the positions of self-agent observations $o^1$ tokens. The modality-specific linear layers for the self-agent observations $o^1$, opponents actions $a^{-1}$, self-agent actions $a^1$, and self-agent rewards $r^1$ comprise 32 nodes without activation functions. All the output heads use linear layers. The output head of $\check{Q}_{\text{C}}$ uses Sigmoid as activation functions.

- *Input & Output*: Given timestep $t$, the input sequences for the Encoder and the Decoder are $(o_1^{-1}, a_1^{-1}, \ldots, o_M^{-1}, a_M^{-1})$ and $(o_{t-L+1}^1, a_{t-L+1}^{-1}, a_{t-L+1}^1, r_{t-L+1}^1, \ldots, o_{t-1}^1, a_{t-1}^{-1}, a_{t-1}^1, r_{t-1}^1, o_t^1)$, respectively. Within, $L$ is the *in-episode historical trajectory sequence length* for the Decoder, as Transformer model has a token capacity. The output prediction sequence is $(\pi_{\text{OI}(t-L+1)}, \check{Q}_{\text{C}(t-L+1)}^h, \check{Q}_{\text{V}(t-L+1)}^h, \ldots, \pi_{\text{OI}(t)}, \check{Q}_{\text{C}(t)}^h, \check{Q}_{\text{V}(t)}^h)$. During training, the corresponding output label sequence is $(a_{t-L+1}^{-1}, \{\mathbb{1}\{r_{t'}^1 \neq 0\}\}_{t'=t-L+1}^{t-L+H}, \{r_{t'}^1\}_{t'=t-L+1}^{t-L+H}, \ldots, a_t^{-1}, \{\mathbb{1}\{r_{t'}^1 \neq 0\}\}_{t'=t}^{t+H-1}, \{r_{t'}^1\}_{t'=t}^{t+H-1})$.

## E.2. Neural Architectures of OOM Baselines

For all the **OOM baselines**, we strictly follow the neural architecture designs adopted in Jing et al. (2024a):

- **DRON-concat**: We use the GPT2 model mentioned in Sec. E.1 to implement the backbones of DRON-concat. DRON-concat's neural architecture contains 2 backbones, and the backbones' architectural design is identical to that of Truncated Q's Decoder, except that it does not include the cross-attention layer. One of the backbones is used to encode the sequence consisting of RTGs $G^1$, observations $o^1$, and actions $a^1$, while the other backbone is used to encode the opponent's *hand-crafted features* (He et al., 2016a). To be versatile in different environments, we consider using the opponent's previous action and the opponent's previous action frequency as the hand-crafted features. We concatenate the embedding tokens obtained by the two backbone encodings in the dimension of the hidden state and feed them to a *fusion linear layer* to get *concatenated embedding tokens*. Within, the *hand-crafted feature embedding tokens* are aligned with the positions of $a^1$ embedding tokens, and other positions are filled with 0 and aligned with the positions of $G^1, a^1$ embedding tokens. Finally, given the concatenated embedding tokens, the actions $a_t^1$ are autoregressively predicted at the locations of observations $o_t^1$. Both the fusion linear layer and the linear layer of hand-crafted features comprise 32 nodes without activation functions, while the number of other hidden layer nodes and the activation function settings are identical to those of Truncated Q's Decoder. All the output heads use linear layers.

- **DRON-MoE**: We use the GPT2 model mentioned in Sec. E.1 to implement the backbones of DRON-MoE. DRON-MoE's neural architecture contains 2 backbones, and the backbones' architectural design is identical to that of Truncated Q's Decoder, except that it does not include the cross-attention layer. One backbone is used to encode the sequence consisting of RTGs $G^1$, observations $o^1$, and actions $a^1$, while the other backbone is used to encode $G^1, o^1, a^1$ sequence and the opponent's hand-crafted features. For the first backbone: After encoding $G_t^1, o_t^1, a_t^1$, the $o_t^1$ embedding tokens are fed to an *expert linear layer* and output a predefined number of expert of action probability distributions. For the second backbone: Before inputting, the opponent's hand-crafted features are concatenated with the corresponding observations $o^1$ according to the timestep, and they are fed to a *mixing linear layer*; after encoding $G_t^1, o_t^1, a_t^1$ and hand-crafted features, $o^1$ embedding tokens are fed to an *expert linear layer*, and a weight vector with a length of the number of experts is outputted. Finally, the obtained weight vector is used to perform a weighted summation of the obtained action probability distributions, and the actions $a_t^1$ are predicted autoregressively. The mixing linear layer consists of 32 nodes without activation function, and the expert linear layer consists of 32 nodes with Softmax (Bridle, 1989) activation function. The number of other hidden layer nodes and the activation function settings are identical to Truncated Q's Decoder. All the output heads use linear layers. The predefined number of experts is set to 5.

- **LIAM**: We use the GPT2 model mentioned in Sec. E.1 to implement the backbones of LIAM. The architectural design of LIAM's backbone is the same as that of Truncated Q's Decoder, except that it does not include a cross-attention layer. In addition to feeding the sequence consisting of RTGs $G^1$, observations $o^1$, and actions $a^1$ into the backbone and predicting the action $a^1$ in an autoregressive manner using a causal self-attention mask, we also employ an *extra decoder* to learn an auxiliary task (Papoudakis et al., 2021a). This auxiliary task involves reconstructing the opponent's observations $o_t^{-1}$ and actions $a_t^{-1}$ using the observations $o_t^1$ and actions $a_{t-1}^1$ of the self-agent. Specifically, we use the $o_t^1$ token embeddings obtained through the backbone as input to the extra decoder to predict $o_t^{-1}$ and $a_t^{-1}$, as the $o_t^1$ token embeddings contain all the information of $o_t^1$ and $a_{t-1}^1$. The extra decoder consists of two linear layers with 32

nodes and no activation functions. The number of other hidden layer nodes and the activation function settings are identical to Truncated Q's Decoder. All the output heads use linear layers.

- **Prompt-DT**: We implement Prompt-DT's neural architecture directly based on its open-source code, which is available at `https://github.com/mxu34/prompt-dt`. We use *high-quality data* against opponent policies from offline datasets as expert demonstrations (*i.e.*, *prompts*). High-quality data refers to trajectories of the self-agent that rank near the top in terms of *return* across all episodes in which it competes against a specific opponent policy. Specifically, we select the top $20\%$ for the high-quality data, sample 3 trajectory, and then sample consecutive fragments with a length of 5 from the trajectory as prompts. The linear layers for RTGs, observations, and actions in the prompts consist of 32 nodes without activation functions. The number of other hidden layer nodes and the activation function settings are identical to Truncated Q's Decoder. All the output heads use linear layers.

- **TAO**: We implement TAO's neural architecture directly based on its open-source code, which is available at `[TAO Code]`. (1) *TAO's Encoder*: We implement TAO's encoder using the exact same backbone as the Encoder of Truncated Q. The opponents' actions $a^{-1}$, rewards $r^{-1}$, and observations $o^{-1}$ are fed into modality-specific linear layers, and a positional episodic timestep encoding is added. We use a *fusion linear layer* to fuse the $a_{t-1}^{-1}, r_{t-1}^{-1}, o_t^{-1}$ embedding tokens at each timestep into *fused embedding tokens*. The sequences of fused embedding tokens are fed into the backbone, which autoregressively outputs the *per-timestep embedding tokens* corresponding to each $(a_{t-1}^{-1}, r_{t-1}^{-1}, o_t^{-1})$ tuple using a causal self-attention mask. Then, we input all the per-timestep embedding tokens output by the backbone, as **key** and **value** into the cross-attention layers of TAO's decoder. The modality-specific linear layers for the opponents' actions, rewards, and observations are composed of 32 nodes with ELU (Clevert et al., 2015) activation functions. (2) *TAO's Decoder*: We implement TAO's decoder using the exact same backbone as the Decoder of Truncated Q. The self-agent's RTGs $G^1$, observations $o^1$, and actions $a^1$ are fed into modality-specific linear layers, and a positional episodic timestep encoding is added. The sequences consisting of embedding tokens of $G_t^1, o_t^1, a_t^1$ are fed into the cross-attention layers of the backbone as **query**, autoregressively predicts actions $a_t^1$ at the positions of $o_t^1$ embedding tokens using a causal self-attention mask. The modality-specific linear layers for the self-agent's RTGs, observations, and actions are composed of 32 nodes without activation functions. All the output heads use linear layers.

# F. Hyperparameters

## F.1. Hyperparameters for Opponent Policies & Offline Datasets

| Hyperparameter Name | MS | LBF | PD | PP |
|---|---|---|---|---|
| Dimensionality of observations | 12 | 21 | 10 | 16 |
| Dimensionality of actions | 5 | 6 | 5 | 5 |
| Horizon for each episode ($T$) | 50 | 50 | 100 | 100 |
| Agent index of the self-agent | 0 | 0 | 0 | 3 |
| Agent indexes of the opponents | 1 | 1 | $1, 2$ | $0, 1, 2$ |
| Size of MEP population | 4 | 4 | 4 | 4 |
| Weighting coefficient for MEP's diversity objective | $1e-2$ | $1e-2$ | $1e-2$ | $1e-2$ |
| Total number of episodes used for training MEP population | $1e5$ | $1e5$ | $1e5$ | $1e5$ |
| Total number of episodes used for training *self-agent policies* (these policies are used to construct offline datasets $\mathbb{T}$, the training objective is to achieve the BR against the opponent) | $1e4$ | $1e4$ | $1e4$ | $1e4$ |
| Number of updating epochs at each MEP's and self-agent policies' training step | 10 | 10 | 10 | 10 |
| Batch size for training MEP population and self-agent policies | 4096 | 4096 | 4096 | 4096 |
| Learning rate for training MEP population | $5e-4$ | $5e-4$ | $5e-4$ | $5e-4$ |
| Learning rate for training self-agent policies | $8e-5$ | $8e-5$ | $8e-5$ | $8e-5$ |
| Maximum norm of the gradients for training MEP population and self-agent policies | 5.0 | 5.0 | 5.0 | 5.0 |
| Clipping factor of PPO (Schulman et al., 2017) used for training MEP population and self-agent policies | 0.2 | 0.2 | 0.2 | 0.2 |
| Number of opponent policies used to construct the offline dataset $\mathbb{T}$ ($K$) | 12 | 12 | 12 | 12 |
| Number of trajectories used to construct the $\mathbb{T}^k$ for each opponent policy $\pi^{-1,k}$ | $1e3$ | $1e3$ | $1e3$ | $1e3$ |

### F.2. Hyperparameters for OOM Baselines Pretraining

| Hyperparameter Name | MS | LBF | PD | PP |
|---|---|---|---|---|
| In-episode historical trajectory sequence length for the backbone (see Sec. E.2 for detailed descriptions) | 20 | 20 | 20 | 20 |
| Sequence length of the In-Context Data $D$ ($M$) | 15 | 15 | 15 | 15 |
| Number of trajectories sampled to constrcut $D$ ($C$) | 3 | 3 | 3 | 3 |
| Total number of training steps | $3e3$ | $3e3$ | $3e3$ | $3e3$ |
| Number of updating epochs at each training step | 10 | 10 | 10 | 10 |
| Batch size | 20 | 20 | 20 | 20 |
| Warm-up epochs (the learning rate is multiplied by $\frac{num\_epoch+1}{warm\text{-}up\ epochs}$ to allow it to increase linearly during the initial warm-up epochs of training) | $1e4$ | $1e4$ | $1e4$ | $1e4$ |
| Reward scaling factor (all the rewards are multiplied by $\frac{1}{reward\ scaling\ factor}$ to reduce the variance of training) | 1 | 1 | 100 | 100 |
| Learning rate for AdamW (Loshchilov & Hutter, 2018) optimizer | $1e-4$ | $1e-4$ | $1e-4$ | $1e-4$ |
| Weight decay coefficient for AdamW optimizer | $1e-4$ | $1e-4$ | $1e-4$ | $1e-4$ |
| Maximum norm of the gradients (clip if exceeded) | 0.5 | 0.5 | 0.5 | 0.5 |
| Dropout factor for the backbone (GPT2 model) | 0.1 | 0.1 | 0.1 | 0.1 |
| Random seeds | $0,1,2,3,4$ | $0,1,2,3,4$ | $0,1,2,3,4$ | $0,1,2,3,4$ |

### F.3. Hyperparameters for Truncated Q Training

| Hyperparameter Name | MS | LBF | PD | PP |
|---|---|---|---|---|
| In-episode historical trajectory sequence length for the Decoder ($L$) (see Sec. E.1 for detailed descriptions) | 20 | 20 | 20 | 20 |
| Sequence length of the In-Context Data $D$ ($M$) | 15 | 15 | 15 | 15 |
| Number of trajectories sampled to constrcut $D$ ($C$) | 3 | 3 | 3 | 3 |
| Total number of training steps | $1e4$ | $1e4$ | $1e4$ | $1e4$ |
| Number of updating epochs at each training step | 10 | 10 | 10 | 10 |
| Batch size | 20 | 20 | 20 | 20 |
| Weighting coefficient for *confidence loss* $\mathcal{L}_{Q_c}$ ($\alpha$) | 1.0 | 1.0 | 1.0 | 1.0 |
| Weighting coefficient for *value loss* $\mathcal{L}_{Q_v}$ ($\beta$) | 1.0 | 1.0 | 1.0 | 1.0 |
| Warm-up epochs | $1e4$ | $1e4$ | $1e4$ | $1e4$ |
| Learning rate for AdamW optimizer | $5e-3$ | $5e-3$ | $5e-3$ | $5e-3$ |
| Weight decay coefficient for AdamW optimizer | $1e-4$ | $1e-4$ | $1e-4$ | $1e-4$ |
| Maximum norm of the gradients (clip if exceeded) | 0.5 | 0.5 | 0.5 | 0.5 |
| Dropout factor for the backbone (GPT2 model) | 0.1 | 0.1 | 0.1 | 0.1 |
| Random seeds | $0,1,2,3,4$ | $0,1,2,3,4$ | $0,1,2,3,4$ | $0,1,2,3,4$ |

## F.4. Hyperparameters for Instant Policy Refinement

| Hyperparameter Name | MS | LBF | PD | PP |
|---|---|---|---|---|
| Truncated horizon ($H$) | 3 | 3 | 3 | 3 |
| Total number of episodes for testing | 2400 | 2400 | 2400 | 2400 |
| Number of episodes between each opponent's non-stationary policy switching | 20 | 20 | 20 | 20 |
| Sequence length of the In-Context Data $D$ ($M$) | 15 | 15 | 15 | 15 |
| Number of historical trajectories of the opponent sampled to construct $D$ ($C$) | 3 | 3 | 3 | 3 |
| Random seeds | $0, 1, 2, 3, 4$ | $0, 1, 2, 3, 4$ | $0, 1, 2, 3, 4$ | $0, 1, 2, 3, 4$ |

