# OpenReview forum: "Offline Opponent Modeling with Truncated Q-driven Instant Policy Refinement"
_ICML.cc/2025/Conference — ICML 2025 poster_

### Official Review · Reviewer_RP82 · 2025-03-10

**Overall Recommendation:** 3

**Summary:**

This paper proposes to learn a horizon-truncated incontext action-value function and a policy refinement mechanism to tackle the offline opponent modeling problem, espically when the training dataset is suboptimal. This paper has  analyzed the rationale of Truncated Q from the perspective of No Maximization Bias probability theoretically. And experimental results have demonstrated the effective of the algorithm.

**Claims And Evidence:**

Yes, the claims made in the submission are supported by both theory and experiment results.

**Essential References Not Discussed:**

I am not very familiar with this field, so I am not sure.

**Experimental Designs Or Analyses:**

Experimental designs are good.

**Methods And Evaluation Criteria:**

The evaluation criteria is make sense, but the proposed algorithm needs more explanation. Please refer to the Question part.

**Other Comments Or Suggestions:**

No

**Other Strengths And Weaknesses:**

Strength:
1. The paper introduces truncated Q-value estimation, which is a interesting approach.
2. This paper provides experimental validation and theoretical analysis, which can serve as an inspiration for future research.
3. The paper is well-structured.

Weakness:
1. Each symbol used in the paper should be clearly defined. The main paper should be self-contained.
2. The pseudocode of the algorithm is necessary in the main part.
3. More intuitive explanation is needed as to why truncating Q-values is better.

**Questions For Authors:**

1. Can you give a detailed explanation about why the error in the optimization objective of $\bar{Q}$ accumulates primarily through the accumulation of rewards over time $t'$ in $G_t^1$ in Equation 1.

2. How to find the optimal $h^*$? Because experimental results show that truncating Q-values performs worse in certain values of h.

3. Intuitively, h should not be too large or too small, and It should have robust performance within a certain range. However, this paper indicates that TIPR’s improvement on OOM algorithm does not exhibit a strict correlation trend with the truncated horizons. This is somewhat counterintuitive. What role does the randomness of hyperparameter tuning experiments play in the experimental results?
.

**Relation To Broader Scientific Literature:**

The key contributions of this paper build upon and extend several important lines of research in offline opponent modeling (OOM), offline reinforcement learning (RL), and in-context learning.

**Theoretical Claims:**

I have reviewed the theoretical section of the paper, but I have some confusions. For example:
1. What's the defination of $\breve{Q}_h$ in Equation 2.
2. The explanations of the notations in Theorem 3.1 should be included in the main paper

---

> ### Author Rebuttal · Authors · 2025-03-28
>
> Thank you for your valuable feedback. In response, we offer the following clarifications and hope our explanations address your concerns and strengthen our work.
>
> ---
>
> > **(A) Theoretical Claims:** 1. What's the definition of $\breve{Q} _\mathsf{h}$ in Equation 2.
> >
>
> $\mathsf{h}$ denotes the random variable of the truncated horizon, and $\mathsf{G} _\mathsf{h}(t) := \sum _{t'=t}^{t+\mathsf{h}-1} \gamma^{t'-t} \mathsf{r} _{t'}$ denotes the random variable of the horizon-truncated *Return-To-Go* (RTG). In Eq. (2), $\breve{Q} _\mathsf{h}$ denotes the random variable corresponding to the Truncated Q (i.e., the neural network we aim to learn), and its learning objective is to approximate $\mathbb{E}\mathsf{G} _\mathsf{h}$.
>
> > **(B) Theoretical Claims:** 2. The explanations of the notations in Theorem 3.1 should be included in the main paper
> Weakness: 1. Each symbol used in the paper should be clearly defined …
> >
>
> We create a notation table in [this link](https://ibb.co/m5QGry8L) to provide detailed explanations of all the notations used in Thm. 3.1. These definitions will be incorporated into the main text in our revision.
>
> > **(C)** Weakness: 2. The pseudocode of the algorithm is necessary in the main part.
> >
>
> We provide the pseudocode for the TIPR algorithm framework at [this link](https://ibb.co/9H8Hjwt8). We appreciate your suggestion and will incorporate it into the main text in our revision.
>
> > **(D)** Weakness: 3. More intuitive explanation is needed as to why truncating Q-values is better.
> >
>
> For this question, please refer to our response to reviewer **wBej**'s comment "**(B) Methods And Evaluation Criteria:** … However, it is still unclear". You can use Ctrl + F to search for the sentence in quotation marks to quickly jump to there.
>
> > **(E) Questions For Authors:** 1. Can you give a detailed explanation about why the error in the optimization objective of $\bar{Q}$ accumulates …
> >
>
> In Eq. (1), the original Q-value $\bar{Q}$ regresses onto the RTG label $G_t^1$. As the horizon grows longer, estimation errors for individual rewards accumulate due to the cumulative summation of discounted rewards. This compounding of errors directly reduces the empirical accuracy of the learned $\bar{Q}$, especially when datasets are suboptimal.
>
> From a theoretical view (Thm. 3.1), this accumulation manifests explicitly in the term $\Delta(\mathsf{h})$, whose complexity increases with the $\mathsf{h}$ (specifically $\Delta(\mathsf{h})=O(\mathsf{h}^{2})$). Consequently, as the $\mathsf{h}$ extends, the Empirical Risk NMB Probability (the probability of correctly fitting truncated returns) decreases significantly due to this error accumulation.
>
> The longer horizon in $\bar{Q}$ reduces its reliability in identifying actions with the highest true returns, lowering the Overall NMB Probability. Truncating the horizon limits the error accumulation, improving action selection accuracy and policy refinement—especially in challenging suboptimal dataset settings.
>
> > **(F) Questions For Authors:** 2. How to find the optimal $H^{\ast}$? …
> >
>
> Our primary focus in this work is to demonstrate, both theoretically and empirically, that **just truncating the horizon for Q-value estimation, without careful optimization of $H$, can already provide significant advantages over not truncating at all**. Theoretical insights from Prop. 3.2 explicitly show the existence of an optimal $H^{\ast}$, validating the conceptual soundness of truncation.
>
> Identifying the precise $H^{\ast}$ is indeed an important direction for future research. Here are some possible methods:
>
> 1. **Cross-Validation on the Dataset**: Empirically selecting $H^{\ast}$ by performing cross-validation using subsets of the dataset to minimize prediction errors or maximize validation performance.
> 2. **Adaptive Methods**: This approach adaptively adjusts $H$ during training or testing based on real-time confidence or uncertainty estimates. For example, an adaptive strategy could start with a small horizon and increase it if confidence stays high, or reduce it when uncertainty rises—balancing accuracy and computation dynamically.
>
> > **(G) Questions For Authors:** 3. Intuitively, h should not be too large or too small, and It should have robust performance within a certain range …
> >
>
> In fact, the results in Fig. 5 (**Question** 5 in Section 4.2) clearly demonstrate that setting $H$ too large (e.g., $H = T$, which degenerates into the Original Q) or too small (e.g., $H = 1$) is suboptimal. A moderate value of $h$ tends to yield better performance.
>
> We did not perform an exhaustive search for the optimal $H^{\ast}$ in each environment. In Fig. 3, we used a moderate, untuned $H$. Still, Truncated Q effectively improves suboptimal original OOM policies over the Original Q.
>
> ---
>
> Your comments have greatly helped improve our manuscript. If you have further feedback, we'll address it promptly. If your concerns are resolved, we hope you'll reconsider your rating.

---

### Official Review · Reviewer_LtaJ · 2025-03-14

**Overall Recommendation:** 3

**Summary:**

The paper proposes an offline opponent modeling approach that enhances the consistency of Q-functions by truncating the horizon length during Q-learning, incorporates in-context learning to mitigate distribution shift caused by sub-optimality in offline datasets, and employs test-time policy refinement to further improve policy performance. Experimental results across four tasks demonstrate the effectiveness of the proposed TIPR method and its key components.

**Claims And Evidence:**

Yes. The claims are reasonable and seem technically sound.

**Essential References Not Discussed:**

I have not found yet.

**Experimental Designs Or Analyses:**

Yes. I checked the designs and analyses. Most of the experimental designs are reasonable.

**Methods And Evaluation Criteria:**

Yes.

**Other Comments Or Suggestions:**

1. The experiments may conduct on some larger-scale environments.

2. The MSE comparisons between $\bar{Q}$ and $\bar{Q}_V$ may be exacerbated. As the original Q is used for a longer horizon, thus a normalization may be needed.

**Other Strengths And Weaknesses:**

**Strengths:**
This work integrates truncated Q-learning, in-context learning, and policy refinement to enhance opponent modeling in multi-agent reinforcement learning. TIPR serves as a plug-and-play framework compatible with many offline opponent modeling (OOM) methods, making it a promising tool for broader applications. The experimental design and analysis validated the effectiveness of TIPR.

**Weaknesses:**

1. The authors use a MEP approach to generate 4 opponents across all four environments, which possibly cover the opponent policy spaces. While this may be sufficient for the current task, it raises concerns about whether the results generalize to larger scenarios.
2. The in-context learning module uses a dataset of sequence length 15 across four experiments. Given that Transformer architectures typically require large-scale pre-training data, the scalability of TIPR in complex environments (e.g., Google Research Football (GRF)) remains unverified.

**Questions For Authors:**

1. How to read the colored and shaded areas of Fig.3 in MS and PP? The shared areas show a permanence drop, which may contradict the legend.
2. Why are the refine condition (RC) threshold and the inner cumulative confidence threshold (indicator condition) set as 0?
3. If I understand correctly, the context length to identify the opponent's newest policy is 15 which is very fast with Transformer (assumed the opponent is seen or unseen but fixed during 20 episodes). So I wonder how  the differences between the trained opponents are.
4. The Appendix E.1 states that the size of MEP population is 4. How are 20 policies sampled from this population for Seen and Unseen setting?
5. Does TIPR introduce significant additional computational overhead (in terms of time and memory) during testing?

**Relation To Broader Scientific Literature:**

This work provides an opponent modeling based approach for MARL.

**Theoretical Claims:**

Yes. I just roughly checked the proof in the appendix and had not found any issues.

---

> ### Author Rebuttal · Authors · 2025-03-28
>
> Thank you for your valuable feedback. We'd like to offer the following clarifications and hope our responses address your concerns and strengthen our work.
>
> ---
>
> > **(A) Weaknesses:** 1. The authors use a MEP approach to generate 4 opponents …
> **Questions For Authors:** 4. … How are 20 policies sampled …
> >
>
> MEP uses Population-Based Training (PBT), which maintains a fixed population of 4 policies and evolves them through cross-play and natural selection. During training, we regularly saved checkpoints of all policies in the population, resulting in many policies. We then selected 20 diverse, representative ones (from weak to strong) to form the opponent policy pool.
>
> > **(B) Weaknesses:** 2 … the scalability of TIPR in complex environments (e.g., Google Research Football (GRF)) remains unverified.
> >
>
> Regarding scalability, we argue that **TIPR framework is fundamentally built upon highly scalable in-context learning methods and the Transformer architecture**, which perfectly aligns with the principles outlined in Richard S. Sutton’s "*The Bitter Lesson*" [1], emphasizing the inherent scalability of learning-based approaches.
>
> Regarding the choice of environments, although they are not highly complex, existing OOM algorithms still exhibit significant performance drops when the datasets are suboptimal. Moreover, these environments are widely recognized as benchmarks in the OM community and have been extensively used in prior works such as DRON, LIAM, TAO, OMIS [2], etc. While environments like GRF and SMACv2 are indeed more complex, they are typically used in MARL, focusing on agent communication and coordination, not the OM setting.
>
> To further validate TIPR, we also ran preliminary experiments on the more challenging **OverCooked** (OC) environment [2,3], which has high-dimensional image inputs and requires intensive cooperation to complete a sequence of subtasks for serving dishes. The results can be found at [this link](https://ibb.co/XZX5X1m9). Due to time limits, we currently report results only on the representative OOM algorithm TAO, and we will include the full results in the revision. TIPR consistently improves the original OOM policy in OC, effectively addressing the suboptimality of datasets.
>
> [1] http://www.incompleteideas.net/IncIdeas/BitterLesson.html?ref=blog.heim.xyz, Rich Sutton.
>
> [2] Opponent modeling with in-context search, NeurIPS2024.
>
> [3] On the Utility of Learning about Humans for Human-AI Coordination, NeurIPS2019.
>
> > **(C)  Other Comments Or Suggestions:** 2. The MSE comparisons between $\bar{Q}$ and $\bar{Q}_V$ may be exacerbated …
> >
>
> Our main concern is the absolute Q-value error during Policy Refinement (PR) in unknown test environments. PR relies on the ranking of Q-values across all legal actions, since actions are chosen based on these rankings. Normalization doesn’t affect this ranking—it only scales values uniformly. Therefore, the reported MSE directly reflects confidence in action rankings for each method. Normalization wouldn't change qualitative conclusion that truncated Q-values improve ranking accuracy and thus enhance PR effectiveness.
>
> > **(D) Questions For Authors:** 1. How to read the colored and shaded areas of Fig.3 in MS and PP? …
> >
>
> We use ***Black Error Bars*** (BEB) for the original OOM policy's Standard Deviation (SD) and *Grey Error Bars* (GEB) for the SD after applying TIPR. In both MS and PP, GEB consistently appears above BEB, with the shaded area showing performance gains. This shows TIPR consistently improves OOM performance. We'll refine Fig. 3 in the revision to address any clarity issues.
>
> > **(E) Questions For Authors:** 2. Why are the refine condition (RC) threshold and … set as 0?
> >
>
> This design is intuitive: (1) **Indicator Condition**: If a legal action is predicted to yield non-zero reward within $H$ steps, its value estimate is considered high-confidence. (2) **RC**: If any action meets this, we deem it promising to use Truncated Q to refine the policy at that timestep. This helps in sparse-reward environments and doesn't negatively impact performance in dense-reward ones.
>
> > **(F) Questions For Authors:** 3 … So I wonder how the differences between the trained opponents are.
> >
>
> Regarding the diversity of the opponent policies, please refer to our response to reviewer **Chog**'s comment "**(D) Experimental Designs Or Analyses:** … However, if the offline dataset". You can use Ctrl + F to search for the sentence in quotation marks to quickly jump to there.
>
> > **(G) Questions For Authors:** 5. Does TIPR introduce significant additional computational overhead …?
> >
>
> For this question, please refer to our response to reviewer **Chog**'s comment "**(G) Other Strengths And Weaknesses:** 2. The computational efficiency".
>
> ---
>
> Your comments have greatly improved our paper. If you have further feedback, we'll address it promptly. If your concerns are resolved, we hope you'll reconsider your rating.

---

> > ### Comment · Reviewer_LtaJ · 2025-04-08
> >
> > Thanks for the clarification.

---

### Official Review · Reviewer_wBej · 2025-03-14

**Overall Recommendation:** 3

**Summary:**

This work aims to model opponent in multi-agent learning scenario from offline data. They argue that the offline data may be suboptimal and lead to suboptimal policies. To address this, they propose to learn a horizon-truncated in-context Q-function and whether to perform policy refinement. Truncated Q-driven Instant Policy Refinement (TIPR) maximize the No Maximization Bias (NMB) probability.

**Claims And Evidence:**

They present somewhat theoretical and experimental results to validate their claim. They provide results by isolating the performance improvement of truncated Q and iterative policy refinement across different testbeds. The experimentation includes different rational ablations.

**Essential References Not Discussed:**

N/A

**Experimental Designs Or Analyses:**

The experiments are well-designed and throughly conducted.

**Methods And Evaluation Criteria:**

The proposed method appears to perform better in terms of experimental evaluation. However, it is still unclear to me why and how truncated Q value would address the sub-optimality of the dataset. I understand truncated Q may reduce the complexity that arise with the increment of the number of opponents.

**Other Comments Or Suggestions:**

The paper should define the concept "policy refinement" in a more concrete way as it is central to the paper.

**Other Strengths And Weaknesses:**

The results show that the value of fixed horizon doesn't impact the performance. It is not clear why this is the case. Also, it would be nice to see whether there is any difference between using lower value of h vs. values close to T.

**Questions For Authors:**

Should we consider the method as few-shot adaptation method? It seems during test time it observes few samples first and then decide whether to perform policy refinement.

**Relation To Broader Scientific Literature:**

This paper nicely fits in between offline MARL, in-context learning, and transformer-based decision making models. However, it discusses a very specific problem opponent modeling via offline pre-training particularly at the presence of suboptimal data.

**Theoretical Claims:**

The proofs looks reasonable. However, as mentioned in the previous point, I am concerned about how truncated Q is a sufficient solution.

---

> ### Author Rebuttal · Authors · 2025-03-28
>
> Thank you very much for your valuable feedback. In response to your comments, we would like to make the following clarifications and feedback. We hope our explanations and analyses can eliminate concerns and make you find our work stronger.
>
> ---
>
> > **(A) Other Comments Or Suggestions:** The paper should define the concept "policy refinement" in a more concrete way …
> >
>
> In *Offline Opponent Modeling* (OOM), existing algorithms typically assume that the offline dataset is optimal. We refer to the policy trained using these existing algorithms directly on the dataset as the **original policy**. When the dataset is suboptimal, we find that the performance of the original policy drops significantly. Our work aims to design a plug-and-play algorithmic framework to improve the performance of the original policy obtained through OOM algorithms—this is the definition of "**Policy Refinement**" (PR). We will clarify this in our revision.
>
> Notably, our work introduces the novel idea of performing PR instantly at test time, referred to as *Instant Policy Refinement* (IPR), rather than refining the original policy during the offline stage, as done in traditional *Offline Conservative Learning* (OCL) methods. As pointed out by Reviewer **Chog**, this helps mitigate distribution shift, and the advantage of IPR over OCL is also demonstrated in Fig. 4 (**Question 2** in Section 4.2).
>
> > **(B) Methods And Evaluation Criteria:** … However, it is still unclear to me why and how truncated Q value would address the sub-optimality of the dataset …
> >
>
> Inspired by OCL methods, we aim to learn a Q-function to refine the original policy obtained through OOM algorithms. However, learning a workable Q-function in multi-agent environments is highly challenging, as the error in return prediction accumulates significantly when learning an Original Q (see Fig. 1). As you noted, this issue becomes more severe with a larger number of opponents—a claim that is also supported by our Theorem 3.1. In contrast, **our Truncated Q can effectively reduce this accumulation, making it possible to distinguish between optimal and suboptimal actions**.
>
> Our in-depth theoretical analysis (Theorem 3.1 and Proposition 3.2) demonstrates that, under the OOM problem setting, **an optimal truncated horizon exists that maximizes the probability of correctly identifying the best actions—formally, the No Maximization Bias probability**. This theoretical result supports the fact that using Truncated Q for PR is more sound and effective than using the Original Q.
>
> In addition, our proposed IPR method can automatically decide whether to perform PR based on the confidence of action-value estimates from the Truncated Q. It effectively addresses the distribution shift problem commonly found in OCL methods: when uncertain, it falls back on the original policy (conservative), and when confident, it refines the policy (greedy). This trade-off mechanism is validated in Fig. 4 (**Question 4** in Section 4.2) to outperform both always being greedy and always being conservative.
>
> > **(C) Other Strengths And Weaknesses:** The results show that the value of fixed horizon doesn't impact the performance …
> >
>
> In Fig. 5 (**Question 5** in Section 4.2), we analyze how the choice of different truncated horizons $H$ for the Truncated Q affects the improvement results. In fact, the results in Fig. 5 show that different values of $H$ lead to varying levels of improvement, and this effect is environment-dependent.
>
> When $H$ is set to a small value, the improvement from PR is limited. When $H$ equals $T$ (i.e., degenerating to the Original Q), PR generally degrades the performance of the original policy. However, when $H$ takes on a moderate value, PR is able to effectively improve the original policy. This observation supports our Proposition 3.2: there exists an optimal $H^{\ast}$, and in general, $H^{\ast}$ is not equal to $T$.
>
> > **(D) Questions For Authors:** Should we consider the method as few-shot adaptation method? …
> >
>
> You've provided a very interesting insight—indeed, our method can be viewed as a form of few-shot adaptation. This is because our Truncated Q is built upon in-context learning and a Transformer-based architecture, which enables it to collect a small set of contextual samples at test time. Through the IPR method, we can adaptively decide whether to perform PR using the Truncated Q—based on the collected samples—without relying on gradient descent. This allows us to generate a refined policy in an adaptive and efficient manner.
>
> ---
>
> All your questions and feedback have greatly contributed to improving our manuscript. With the valuable input from you and all other reviewers, the quality of our work can be significantly enhanced. We welcome further comments from you and will seriously consider your suggestions for revisions. If you feel that we have addressed your concerns, we hope you will reconsider your rating.

---

### Official Review · Reviewer_Chog · 2025-03-16

**Overall Recommendation:** 3

**Summary:**

This paper introduces a framework called Truncated Q-driven Instant Policy Refinement (TIPR) to improve Offline Opponent Modeling (OOM) algorithms trained on suboptimal datasets where the self-agent may not always select best-response (BR) actions to its opponents. Unlike prior OOM methods that assume optimal trajectories, TIPR addresses the reality that self-agent policies in offline datasets can be arbitrarily poor. The framework introduces Truncated Q, a horizon-truncated in-context action-value function that estimates returns over a fixed truncated horizon rather than the full episode, reducing estimation complexity and improving reliability. At test time, Instant Policy Refinement (IPR) dynamically refines the self-agent policy by using Truncated Q’s confidence estimates to decide whether policy updates are necessary. Theoretical analysis suggests that Truncated Q optimizes the No Maximization Bias (NMB) probability and there exist optimal truncated horizon lengths which can stabilize training along with ensuring effectiveness of Q-guided action selection. Empirical results across four competitive multi-agent environments show that TIPR consistently improves various OOM algorithms, even with highly suboptimal datasets. TIPR also outperforms conventional offline RL methods like Offline Conservative Learning (OCL) which struggle with distributional shifts at test time.

**Claims And Evidence:**

Claim: Suboptimal datasets degrade OOM performance
- The paper presents experimental results across four competitive environments showing that existing OOM algorithms perform poorly when trained on suboptimal datasets. This supports the claim that suboptimality is a significant issue.

Claim: Truncated Q is more reliable than Original Q
- Theoretical analysis explains why truncating the Q function reduces estimation complexity and improves learning stability.
- Experimental results (Figure 1, Table 1) show that Truncated Q has significantly lower Mean Squared Error (MSE) than Original Q, supporting the claim that it provides more reliable value estimates.

Claim: TIPR consistently improves OOM algorithms across varying dataset optimality levels
- Performance improvements in Figure 3 demonstrate that TIPR enhances multiple OOM algorithms, even with heavily suboptimal datasets.
- The paper includes a thorough ablation study, showing that removing key components (confidence estimation, in-context opponent modeling) reduces performance.

Claim: IPR is more effective than OCL for OOM improvement
- Empirical results (Figure 4) show that IPR outperforms OCL across all environments.
- The explanation that OCL struggles due to distributional shifts is reasonable and aligns with existing challenges in offline RL.

Claim: Theoretical justification of Truncated Q maximizing the No Maximization Bias (NMB) probability
- The analysis in Section 3.3 argues that Truncated Q is superior due to better balancing between empirical risk and natural NMB probability.
- However, the derivation relies on assumptions about the structure of Q-learning errors and the properties of the opponent policies, which may not hold universally.
- Empirical validation supports the effectiveness of Truncated Q, but additional experiments explicitly comparing different horizon truncation strategies could strengthen this claim.

Claim: TIPR maintains stable improvements across all degrees of suboptimality
- While the results show consistent performance gains, some environments (PP) exhibit high variance in TIPR's impact (Figure 3). More analysis on why TIPR works better in some settings than others would improve clarity.

Claim: Optimal truncated horizon $h^*$ exists for every environment
- Proposition 3.2 suggests that there is an optimal $h^*$, but finding it is non-trivial. This paper does not provide a method to determine $h$ optimally and instead treats it as a hyperparameter. The claim would be stronger if supported by an adaptive mechanism for choosing $h$ rather than manually testing different values (as in Figure 5).

Overall, the core claims (that TIPR improves OOM algorithms, that Truncated Q is better than Original Q, and that IPR is more effective than OCL) are well-supported by both theory and experiments. However, some theoretical claims (like No Maximization Bias probability, optimal truncation horizon) rely on assumptions that may not always hold and would benefit from additional empirical validation.

**Essential References Not Discussed:**

Related to this paper's discussion on adapting the self-agent to unseen opponents, the problem formulation in [1] focuses on a similar setting from the perspective of principal-agent mechanism design using a meta-RL approach to few-shot test-time adaptation.

[1] Banerjee, A., Phade, S., Ermon, S. and Zheng, S., 2023. MERMAIDE: Learning to Align Learners using Model-Based Meta-Learning. arXiv preprint arXiv:2304.04668.

**Experimental Designs Or Analyses:**

Yes, I checked the entire experiments section in the paper.

This work assumes that the opponent policies seen during training provide enough diversity such that Truncated Q generalizes well to new opponents. However, if the offline dataset lacks sufficient opponent diversity, then the confidence and value estimates of Truncated Q might be biased. This issue is hinted at in Table 1, where accuracy for the confidence estimates $Q_C$ drops in environments with multiple opponents (PP for example). The assumption that an in-context model can fully capture opponent dynamics may not hold if the training distribution is too narrow.

**Methods And Evaluation Criteria:**

Yes

**Other Comments Or Suggestions:**

The last paragraph in page 2 introduces the variable $M$ but has it been defined somewhere in the paper?

**Other Strengths And Weaknesses:**

Additional comments on some limitations in this paper:

1. While the proposed approach is evaluated on smaller simulated environments for multi-agent RL, a discussion on real-world deployment challenges (for example, computational cost of real-time policy refinement) is missing. Testing TIPR on more complex strategic settings (like SMACv2) would better demonstrate its practical impact.

2. The computational efficiency of TIPR is not discussed. Since IPR refines policies dynamically, it may be slower than standard OOM baselines.

**Questions For Authors:**

None

**Relation To Broader Scientific Literature:**

Opponent modeling has traditionally relied on online learning, with early works like Deep Reinforcement Opponent Network (DRON) (He et al., 2016) and Latent Interaction-based Opponent Modeling (LIAM) (Papoudakis et al., 2021) focusing on learning opponent representations. More recent efforts, such as TAO (Jing et al., 2024), leveraged transformers for in-context learning to adapt to unseen opponents. However, previous Offline Opponent Modeling (OOM) approaches assumed that datasets contained optimal trajectories, a limitation that TIPR overcomes by enabling learning from suboptimal offline datasets. The idea of refining policies based on action-value functions is inspired by conservative offline RL methods, such as Conservative Q-learning (CQL) (Kumar et al., 2020) and Implicit Q-learning (IQL) (Kostrikov et al., 2021), but TIPR differs by performing refinement dynamically at test time rather than during offline training. This paper also contributes to the growing field of transformer-based RL, drawing inspiration from Decision Transformer (Chen et al., 2021) and Prompt-DT (Xu et al., 2022), but instead of using ICL for direct policy learning, TIPR applies it to improve action-value estimation through Truncated Q. Additionally, it addresses challenges in multi-agent RL related to opponent non-stationarity by introducing a more flexible confidence-based policy refinement mechanism, allowing real-time adaptation to shifting opponents. However, unlike adaptive truncation methods used in Q-learning (Poiani et al., 2023), TIPR does not yet provide an automated mechanism to determine the optimal truncation horizon $H$.

**Theoretical Claims:**

I have briefly looked at the proof in the appendix. Reiterating the comment made in the "Claims and Evidence" section:

The proof relies on a structured relationship between truncation horizon and error growth, which may not generalize across all settings. Specifically, the empirical risk bound in eq-4 assumes a specific distribution of Q-learning errors, which depends on well-behaved rewards and opponent policies. If the environment dynamics cause non-monotonic error growth (for example due to long-term dependencies in strategic games), the proposed bounds might not accurately reflect reality. Further, the assumption that opponent modeling via in-context data is always reliable could be a strong assumption, especially since some environments may have non-stationary opponents whose strategies evolve in unseen ways.

---

> ### Author Rebuttal · Authors · 2025-03-28
>
> Thank you for your feedback. We'd like to clarify a few points in response, hoping our explanations address your concerns and strengthen our work.
>
> ---
>
> > **(A) Theoretical Claims: …** the empirical risk bound in eq-4 assumes a specific distribution of Q-learning errors, which depends on well-behaved rewards and opponent policies. If the environment dynamics cause non-monotonic error growth …
> >
>
> Our theory (Thm. 3.1 and Prop. 3.2) fundamentally proves that there exists a trade-off between the truncated horizon $h$ and the No Maximization Bias probability. Specifically, the theorem reveals a structured relationship showing that extending the $h$ increases the complexity of accurately estimating returns, reducing the accuracy of Q-values in selecting optimal actions. Thus, using Truncated Q is more sound and effective than using the Original Q for policy refinement.
>
> Importantly, we clarify that **our theory does not explicitly assume or depend on any particular structure or specific distributions of Q-learning errors, nor does it require particular properties or assumptions regarding opponent policies**. The derived bounds in Eq. (4) are general and are based purely on standard statistical inequalities and complexity terms that arise naturally from the estimation problem itself.
>
> We acknowledge that some environment dynamics may cause non-monotonic error growth. However, our theory doesn't specifically assume monotonicity or well-behaved rewards or opponents. Future work could explore more detailed analyses of environments with complex long-term strategic dependencies.
>
> > **(B) Theoretical Claims:** Further, the assumption that opponent modeling via in-context data is always reliable could be a strong assumption …
> >
>
> We clarify that assuming reliable Opponent Modeling (OM) via *In-Context Data* (ICD) is not overly strong. TAO (Jing et al., 2024a) has theoretically and empirically proved that using ICD for OM is both feasible and effective, even with non-stationary opponents. Building on this evidence, our Truncated Q uses ICD to enhance OM reliability and performance.
>
> > **(C) Claims And Evidence:** The claim would be stronger if supported by an adaptive mechanism for choosing $h$ rather than manually testing different values (as in Figure 5).
> >
>
> For possible automated mechanisms of choosing $h$, please refer to our response to reviewer **RP82**'s comment "**(F) Questions For Authors:** 2. How to find". You can use Ctrl + F to search for the sentence in quotation marks to quickly jump to there.
>
> > **(D) Experimental Designs Or Analyses:** … However, if the offline dataset lacks sufficient opponent diversity …
> >
>
> We use MEP algorithm to generate opponent policies. From the MEP population, 12 policies were selected as training opponents (Seen Set), and 8 policies were selected as test opponents in the Unseen Set. **A quantitative analysis of the diversity among the 12 training opponent policies** is provided at [this link](https://ibb.co/XZnpC3cm).
>
> To measure their distinctions, we use the **Pair-wise Expected KL Divergence** (PEKLD). If we take 1.0 as a threshold, over 80% of the PEKLD values in all environments exceed this threshold. This indicates that the training opponent policies are generally well-distinguished from one another.
>
> > **(E) Essential References Not Discussed:** … the problem formulation in [1] focuses on a similar setting from the perspective of principal-agent …
> >
>
> [1] is an interesting related work. We will cite [1] and discuss its connections with our work in our revision.
>
> > **(F) Other Strengths And Weaknesses:** 1 … Testing TIPR on more complex strategic settings …
> >
>
> For experiment results on more challenging settings, please refer to our response to reviewer **LtaJ**'s comment "**(B) Weaknesses:** 2 … the scalability of TIPR".
>
> > **(G) Other Strengths And Weaknesses:** 2. The computational efficiency of TIPR is not discussed …
> >
>
> Compared to the original OOM algorithm, **TIPR only requires one additional forward pass using the Truncated Q at each timestep, which can be completed very quickly (on the order of milliseconds to seconds).**
>
> Many other policy refinement methods, such as decision-time planning (e.g., Monte Carlo Tree Search), perform a large number of forward passes to rollout numerous simulated trajectories to refine the policy. In contrast to them, the computational cost of TIPR is negligible.
>
> > **(H) Other Comments Or Suggestions:** … introduces the variable $M$ but has it been defined somewhere in the paper?
> >
>
> $M$ denotes the number of $(o^{-1}, a^{-1})$ tuples in the ICD $D$. This value determines how much information is used to characterize the opponent's policy and is typically set as a predefined constant. We will include this in our revision.
>
> ---
>
> Your comments have greatly helped improve our paper. If you have further feedback, we're happy to address it. If your concerns are resolved, we hope you'll reconsider your rating.

---

### Decision · Program_Chairs · 2025-05-01

**Decision:**

Accept (poster)

**Comment:**

The paper proposes a framework named Truncated Q-driven Instant Policy Refinement (TIPR) to improve Offline Opponent Modeling (OOM) algorithms trained on suboptimal datasets. Both theoretical analysis and experimental results strongly back the central assertions that TIPR enhances OOM algorithms, Truncated Q outperforms Original Q, and that IPR is more effective than OCL. After the rebuttal discussion, the reviewers consistently think this paper can be accepted.